# TABULAR ANOMALY DETECTION VIA RECONSTRUCTION WITH ATTENTION-BASED BOTTLENECK

## ABSTRACT

Tabular anomaly detection (TAD) is an important task in machine learning, since many real-world datasets are represented in tabular form. However, it remains challenging due to the lack of labeled anomalies and the heterogeneous nature of features. Although many deep learning methods have been developed for TAD, most still rely on simple multilayer perceptrons (MLPs), overlooking architectural design, and in some cases even underperform traditional machine learning methods such as KNN. Motivated by this, we propose LATTE, a simple yet effective reconstruction-based framework that introduces (*i*) an attention-based bottleneck to capture inter-column dependencies and (*ii*) a learnable memory bank, inspired by KNN, to retrieve prototypical normal patterns and amplify anomaly signals. By unifying these components, LATTE consistently outperforms state-of-the-art methods on standard TAD benchmarks.

## 1 INTRODUCTION

*Anomaly detection* is a fundamental task in machine learning, which aims to identify abnormal patterns that deviate from expected behavior. This plays a critical role across a wide range of domains such as finance (Al-Hashedi & Magalingam, 2021), healthcare (Fernando et al., 2021), manufacturing (Esmaeili et al., 2023), and cybersecurity (Malaiya et al., 2019). In particular, *tabular anomaly detection (TAD)* is practically important since a large portion of real-world datasets are naturally represented in tabular form. In many of these applications, however, labeled anomaly datasets are rarely available (Fernando et al., 2021; Al-Hashedi & Magalingam, 2021). As a result, models are often trained under a one-class classification setting (Ruff et al., 2018), where only normal samples are accessible during training. The absence of anomalies, together with the heterogeneous nature and the lack of prior structural knowledge in tabular data, poses significant challenges to effectively modeling abnormal patterns (Grinsztajn et al., 2022; Pang et al., 2021).

To address these challenges, deep learning approaches for TAD have primarily focused on designing learning algorithms in which the loss function is minimized using only normal samples during training, and then the loss is applied at test time to distinguish between normal and abnormal samples. These approaches can be broadly categorized into two groups: (*i*) representation learning methods (Shenkar & Wolf, 2022; Ye et al., 2025a), which model the distribution of normal representations in the latent space (*e.g.*, contrastive learning), and (*ii*) reconstruction-based methods (Yin et al., 2024; Thimonier et al., 2024a), which rely on reconstruction errors measured in the input space. Because capturing intrinsic correlations between input features is particularly important for tabular data, reconstruction-based approaches have been more widely adopted in recent studies.

Despite the development of diverse TAD methods, traditional machine learning approaches such as K-nearest neighbors (KNN) remain strong baselines (Han et al., 2022) and often outperform recent deep learning methods (see Table 1). At the same time, many state-of-the-art reconstruction-based methods still rely on simple multi-layer perceptrons (MLPs) as encoder-decoder architectures. These observations indicate that progress in TAD may require more than revisiting learning algorithms alone; it also calls for greater attention to architectural design, drawing inspiration both (*i*) from how to effectively capture intrinsic correlations between input features, and (*ii*) from why traditional methods like KNN can sometimes achieve superior performance.

**Contribution.** Motivated by the lack of architectural considerations, we propose LATTE, a simple yet effective reconstruction-based TAD framework that leverages an attention-based bottleneck

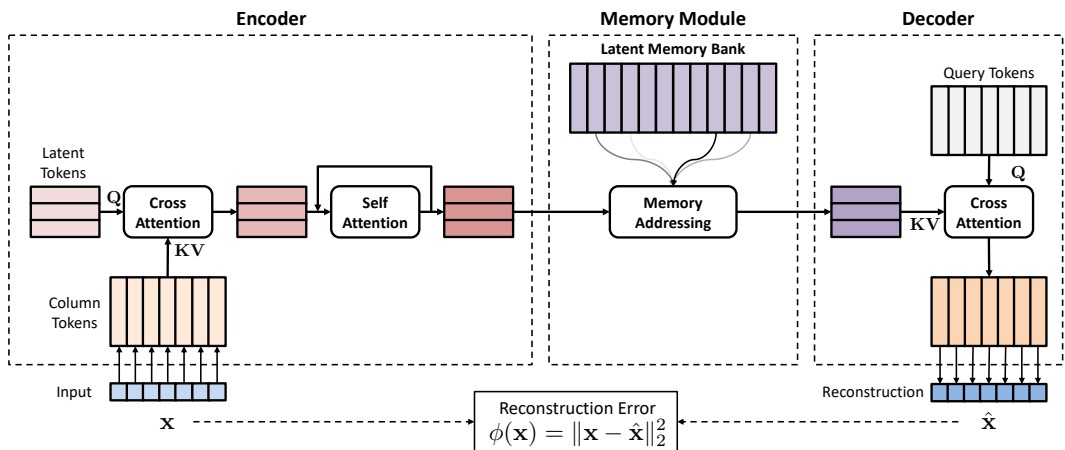

Figure 1: Architecture of our framework.

architecture. Our key idea is to replace the conventional MLP bottleneck with the attention mechanism that dynamically operates on query-key similarities. This allows the model (*i*) to effectively aggregate column information into a set of latent tokens (*i.e.*, bottleneck), and (*ii*) to selectively extract information from these latent tokens to reconstruct each column. In doing so, the model can explicitly model inter-column dependencies that are critical for reliable anomaly detection.

Furthermore, inspired by the strong performance of KNN, we introduce a learnable memory bank that plays a similar role by retrieving prototypical normal patterns. This design projects latent tokens toward representative normal prototypes, thereby amplifying the reconstruction errors of anomalies. Finally, by integrating the attention-based bottleneck with the learnable memory bank, we establish a unified reconstruction-based framework, as illustrated in Figure 1. Similar to prior approaches, the framework is trained by minimizing reconstruction errors on normal samples, and the reconstruction error is used as the anomaly score.

Comprehensive experiments demonstrate that our method achieves superior performance over prior approaches. Specifically, across 20 benchmark datasets and in comparison with 13 recent baselines, our model attains an average AUC-PR of 0.7124 and an average rank of 3.4286, establishing a new state-of-the-art in TAD (see Table 1). In addition, we provide component analysis in Section 4.2 to demonstrate the effectiveness of each part of our model. Beyond this, we provide ablation studies and analysis in Section 4.3 and 4.4, respectively.

## 2 PRELIMINARIES

### 2.1 TABULAR ANOMALY DETECTION

**Problem Statement.** Tabular data consist of multiple *columns*, where each column has a heterogeneous data type such as numerical, categorical, or ordinal. Formally, a tabular data space $\mathcal{X}$ with $F$ columns can be expressed as $\mathcal{X} = \mathcal{C}_1 \times \cdots \times \mathcal{C}_F$ where $\mathcal{C}_i$ is the $i$-th column space. The goal of *tabular anomaly detection (TAD)* is to distinguish between normal and abnormal samples in the tabular data $\mathcal{X}$. In the most common setting (*i.e.*, one-class), the training dataset $\mathcal{D}_{\texttt{train}} = \{\mathbf{x}^{(i)} \in \mathcal{X}\}$ consisting only of normal samples, and the task is to learn a scoring function $\phi : \mathcal{X} \to \mathbb{R}$, where a higher score indicates a higher probability of $\mathbf{x}$ being an anomaly. For evaluation, the test dataset $\mathcal{D}_{\texttt{test}} = \{(\mathbf{x}^{(i)}, y^{(i)}) \in \mathcal{X} \times \{0, 1\}\}$ is provided where $y^{(i)}$ indicates whether $\mathbf{x}^{(i)}$ is normal (*i.e.*, $y^{(i)} = 0$) or anomalous (*i.e.*, $y^{(i)} = 1$). Following recent TAD literature (Yin et al., 2024; Ye et al., 2025b;a), we convert all columns into numerical representations using standard preprocessing techniques such as normalization for numerical features and one-hot encoding for categorical features. Accordingly, throughout this paper, we represent the data space as $\mathcal{X} = \mathbb{R}^F$, which makes the computation of reconstruction errors more straightforward.

**Reconstruction-based TAD.** A widely adopted approach in TAD employs *reconstruction* error as the anomaly scoring function, *i.e.*, $\phi(\mathbf{x}) = \|\mathbf{x} - f(\mathbf{x})\|$, where the autoencoder $f : \mathcal{X} \to \mathcal{X}$ is trained to reconstruct normal samples from $\mathcal{D}_{\texttt{train}}$. The underlying intuition is that normal samples should yield low reconstruction errors because they follow the same distribution as the training data, whereas anomalies, being out-of-distribution, are expected to produce higher errors. Building on this paradigm, several recent methods have proposed refinements. For example, MCM (Yin et al., 2024) introduces a mask generator for reconstruction, while Disent (Ye et al., 2025b) employs a disentanglement technique in the latent space of an autoencoder. However, these methods typically adopt simple multi-layer perceptrons (MLPs) for the autoencoder without exploring architectural designs that might better capture complex inter-feature dependencies in tabular data.

## 2.2 ATTENTION MECHANISM

In this section, we formally define the *attention* mechanism, which serves as a fundamental component of our LATTE architecture. It is designed to capture dependencies between tokens and has been widely applied across diverse domains, including vision (Dosovitskiy et al., 2021), natural language processing (Vaswani et al., 2017), time-series analysis (Nie et al., 2023), and tabular data (Huang et al., 2020). The ability to capture such dependencies is particularly important in tabular anomaly detection, where anomalies often arise from irregular correlations across multiple columns. Formally, the attention mechanism is defined as:

$$\texttt{Attention}(\mathbf{Q}, \mathbf{K}, \mathbf{V}) = \texttt{Softmax}\left(\frac{\mathbf{Q}\mathbf{K}^{\top}}{\sqrt{d}}\right)\mathbf{V} \in \mathbb{R}^{N \times d}, \tag{1}$$

where $\mathbf{Q} \in \mathbb{R}^{N \times d}$, $\mathbf{K} \in \mathbb{R}^{M \times d}$, and $\mathbf{V} \in \mathbb{R}^{M \times d}$ denote the query, key, and value matrices composed of $d$-dimensional token vectors, respectively. A common extension of attention is *multi-head attention (MHA)*, denoted by $\texttt{MHA}(\mathbf{Q}, \mathbf{K}, \mathbf{V})$, which applies multiple attention functions (*i.e.*, heads) in parallel. For simplicity, we omit its learnable parameters (*e.g.*, input and output projections) and the detailed description of MHA.

## 3 METHOD

Motivated by the lack of architectural considerations in the prior literature in tabular anomaly detection (TAD), we propose a simple yet effective reconstruction-based TAD framework, LATTE, that leverages an attention-based bottleneck architecture. Specifically, we first introduce an attention-based autoencoder (Section 3.1), followed by a memory module designed to recover anomalous or perturbed latent representations toward normal ones (Section 3.2). Finally, we present the overall training objective and the anomaly scoring function (Section 3.3). Our overall framework is illustrated in Figure 1.

## 3.1 AUTOENCODER WITH ATTENTION-BASED BOTTLENECK

In reconstruction-based anomaly detection, the autoencoder architecture design is crucial: the bottleneck determines which information is preserved for encoding and decoding, and what information is discarded. However, prior TAD works largely rely on simple MLP encoders and decoders without careful architectural considerations (Shenkar & Wolf, 2022; Yin et al., 2024; Ye et al., 2025a).

Our key idea is to replace this MLP-based bottleneck with a *attention-based latent bottleneck* that effectively aggregates information from the input, extracting essential characteristics of normal samples while filtering out irrelevant details. To this end, we leverage cross-attention layers (*i*) to *compress* the column tokens into latent tokens by weighting columns according to their input-dependent interactions, and (*ii*) to *reconstruct* the input by mapping the latent tokens back to the column space through query-based cross attention with both global and column-specific queries. In what follows, we elaborate the details of (*i*) the *encoder* and (*ii*) the *decoder*.

**Encoder**. Given an input $\mathbf{x} = [x_1, \dots, x_F] \in \mathbb{R}^F$ of $F$ columns as described in Section 2.1, we first embed each column $x_i$ into a column token $\tilde{\mathbf{x}}_i$ in a column-wise manner as follows:

$$\tilde{\mathbf{x}}_i = x_i \mathbf{w}_i + \mathbf{b}_i \in \mathbb{R}^d, \quad i = 1, \dots, F,$$

where $\mathbf{w}_i \in \mathbb{R}^d$ and $\mathbf{b}_i \in \mathbb{R}^d$ are weight and bias vectors for the $i$-th column, respectively. We then denote $\mathbf{X} = [\tilde{\mathbf{x}}_1; \ldots; \tilde{\mathbf{x}}_F] \in \mathbb{R}^{F \times d}$ as the embedding matrix of $F$ column tokens. Note that $\mathbf{b}_i \in \mathbb{R}^d$ can be interpreted as the positional embedding of the $i$-th column. To compress the $F$ column tokens into a smaller number of latent tokens, we employ a cross-attention mechanism. Specifically, we use $M$ learnable latent tokens $\mathbf{Z}_{\mathtt{init}} \in \mathbb{R}^{M \times d}$ for the attention queries as follows:

$$\mathbf{Z} \leftarrow \mathbf{Z}_{\mathtt{init}} + \mathtt{MHA}(\mathbf{Z}_{\mathtt{init}}, \mathbf{X}, \mathbf{X}), \tag{2}$$

$$\mathbf{Z} \leftarrow \mathbf{Z} + \mathtt{FFN}(\mathbf{Z}), \tag{3}$$

where $\mathtt{FFN}(\cdot)$ is a 2-layer feedforward network following the common choice for attention blocks. We here omit layer normalization layers and learnable parameters in $\mathtt{MHA}$ and $\mathtt{FFN}$ for simplicity. This cross-attention enables to compactly aggregate diverse features depending on the input $\mathbf{x}$.

To further enhance the quality of the latent tokens $\mathbf{Z}$, we apply a self-attention layer for capturing inter-latent relationships as follows:

$$\mathbf{Z} \leftarrow \mathbf{Z} + \mathtt{MHA}(\mathbf{Z}, \mathbf{Z}, \mathbf{Z}), \tag{4}$$

$$\mathbf{Z} \leftarrow \mathbf{Z} + \mathtt{FFN}(\mathbf{Z}). \tag{5}$$

We repeat this procedure $L$ times with parameter sharing. The analysis of the effect of this repetitive refinement is provided in Section 4.3. After refinement, the final latent tokens are considered as the output of our encoder, *i.e.*, $\mathbf{Z} = \mathtt{Enc}(\mathbf{x})$. Since we use a small number of latents, *i.e.*, $M \ll F$, the latent tokens $\mathbf{Z}$ act as an effective bottleneck representation.

**Decoder**. Given the latent tokens $\mathbf{Z} = \mathtt{Enc}(\mathbf{x})$, our decoder $\mathtt{Dec}(\cdot)$ reconstructs the original input $\mathbf{x}$ from the latents $\mathbf{Z}$ via cross-attention with column-specific query tokens. Specifically, we construct each query token $\mathbf{q}_i$ for the $i$-th column as follows:

$$\mathbf{q}_i = \mathbf{q}_{\mathtt{global}} + \mathbf{b}_i \in \mathbb{R}^d, \quad i = 1, \ldots, F, \tag{6}$$

where $\mathbf{q}_{\mathtt{global}}$ is a learnable query embedding shared across all the columns, and $\mathbf{b}_i$ is the column-specific positional embedding used in our encoder. We find that combining global and local components yields the most effective performance, validating our proposed query formulation in Section 4.3. We then recover the column tokens through a cross-attention layer as follows:

$$\mathbf{Y} \leftarrow \mathbf{Q} + \mathtt{MHA}(\mathbf{Q}, \mathbf{Z}, \mathbf{Z}), \tag{7}$$

$$\mathbf{Y} \leftarrow \mathbf{Y} + \mathtt{FFN}(\mathbf{Y}), \tag{8}$$

where $\mathbf{Q} = [\mathbf{q}_1; \ldots; \mathbf{q}_F] \in \mathbb{R}^{F \times d}$ is the query embedding matrix, and $\mathbf{Y} = [\mathbf{y}_1; \ldots; \mathbf{y}_F] \in \mathbb{R}^{F \times d}$ is the output column tokens. This procedure enables to extract column-specific information from the latent bottleneck effectively. The final reconstruction is obtained by applying column-specific linear projections as follows:

$$\hat{\mathbf{x}} = [\hat{x}_1, \ldots, \hat{x}_F] \in \mathbb{R}^d, \quad \hat{x}_i = \mathbf{u}_i^\top \mathbf{y}_i + c_i \in \mathbb{R},$$

where $\mathbf{u}_i \in \mathbb{R}^d$ and $c_i \in \mathbb{R}$ are learnable parameters of the linear projections. We simply denote the overall decoding procedure as $\hat{\mathbf{x}} = \mathtt{Dec}(\mathbf{Z})$.

## 3.2 LATENT MEMORY BANK

A key aspect of anomaly detection is to determine abnormality by measuring the relationship between test data and normal samples. This is evidenced by the fact that even classic methods such as $k$-nearest neighbors (KNN) can perform on par with more sophisticated deep learning approaches as shown in Table 1. Motivated by this observation, we introduce a *learnable memory bank* that maintains a set of representative normal patterns in the latent space. Through attention, the encoder latent tokens are replaced by relevant prototypes from the memory bank, effectively mapping anomalous or perturbed latents to normal ones. This design enables our reconstruction-based framework to improve detection capability by amplifying the reconstruction errors of abnormal samples.

We now describe the details of the proposed latent memory bank. This consists of a set of learnable memory vectors that encode prototypical normal patterns, together with an attention-based addressing operator for accessing them. Formally, let $\mathbf{M} = [\mathbf{m}_1; \ldots; \mathbf{m}_K] \in \mathbb{R}^{K \times d}$ be a memory matrix of $K$ learnable memory vectors. Given a latent token $\mathbf{z} \in \mathbb{R}^d$, its memory-enhanced latent is computed

as $\hat{\mathbf{z}} = \sum_{i=1}^{K} w_i \mathbf{m}_i$, where $w_i$ is determined by the cosine similarity between $\mathbf{z}$ and $\mathbf{m}_i$. More generally, this memory addressing can be expressed using the attention formulation as follows:

$$\hat{\mathbf{Z}} = \texttt{Memory}(\mathbf{Z}, \mathbf{M}) = \texttt{Attention}(\texttt{normalize}(\mathbf{Z}), \texttt{normalize}(\mathbf{M}), \mathbf{M}), \qquad (9)$$

where $\texttt{normalize}(\cdot)$ is the row-wise $\ell_2$ normalization operator. In this module, we use the temperature scaling hyperparameter, $\tau$, rather than $\sqrt{d}$ as in Eq. (1). Note that this memory module acts as a weighted retrieval, where $\tau$ controls the sharpness of the attention distribution; for example, when $\tau$ is very small, it operates like 1-NN. More generally, the lookup aggregates multiple memory vectors similar to the input latent $\mathbf{z}$ according to their cosine distances. Therefore, at inference time, our memory module and KNN are functionally analogous in terms of retrieval behavior. We evaluate the sensitivity of the temperature hyperparameter in Section 4.3. As we expected, replacing $\mathbf{Z}$ with $\hat{\mathbf{Z}}$ in this manner is found to improve detection performance, as demonstrated in Section 4.2.

### 3.3 OVERALL FRAMEWORK

In the previous sections, we have introduced the encoder $\texttt{Enc}(\cdot)$, decoder $\texttt{Dec}(\cdot)$, and memory module $\texttt{Memory}(\cdot)$, all designed with attention mechanisms. Given an input $\mathbf{x} \in \mathbb{R}^d$, our reconstruction framework $f$ is formulated as:

$$\hat{\mathbf{x}} = f(\mathbf{x}) = \texttt{Dec}(\texttt{Memory}(\texttt{Enc}(\mathbf{x}), \mathbf{M})). \qquad (10)$$

As described in Section 2.1, the framework $f$ is trained to minimize the reconstruction error, $\|\mathbf{x} - f(\mathbf{x})\|_2^2$, on the training dataset $\mathcal{D}_{\texttt{train}}$ containing only normal samples. We then define the anomaly score $\phi(\cdot)$ as its individual reconstruction error, *i.e.*, $\phi(\mathbf{x}) = \|\mathbf{x} - f(\mathbf{x})\|_2^2$.

## 4 EXPERIMENTS

We design our experiments to answer following questions:

- Does our framework consistently outperform baselines on benchmarks? (§4.1)
- How does individual component contribute to the overall performance? (§4.2)
- Which architecture choices and hyperparameters most improve the performance? (§4.3)
- How does our architecture distinguish between normal and abnormal samples? (§4.4)

**Setup**. Following previous work (Yin et al., 2024), 20 tabular datasets are used for our evaluation. 12 of them are obtained from the Outlier Detection Datasets (Rayana, 2016), while the remainder are derived from ADBench (Han et al., 2022). We randomly sample 50% of the normal samples as the training set, and the remaining normal samples with all anomaly samples are combined into the test set. Area Under the Precision-Recall Curve (AUC-PR) are selected as our evaluation criteria since it is robust to the class imbalance commonly found in TAD (Boyd et al., 2013). This metric can objectively evaluate detection performance without making any assumption on the decision threshold. All reported results are averaged over 10 random seeds. We confirmed that all methods are evaluated on same train-test split on same seeds.

**Implementation Details**. For LATTE, we employed Adam optimizer with an initial learning rate of 1e-3, which exponentially decays at a rate of 0.98, and adopted an early stopping strategy with a patience of 10 and 20 epochs for small-scale and large-scale datasets, respectively. The model was trained with a batch size of 512, except for the Census and Fraud datasets, for which a size of 2048 was used. The model has a self-attention layer repeated $L = 4$ times with an embedding dimension of $d = 64$. The temperature of latent memory bank is set to 0.1. For the number of latent tokens $M$ and the number of learnable memory vectors $K$, we empirically find that setting $M \approx \sqrt{F}$ and $K \approx \sqrt{|\mathcal{D}_{\texttt{train}}|}$ provides a strong default configuration, where $F$ and $|\mathcal{D}_{\texttt{train}}|$ denote the number of columns and train samples, respectively. More details of implementations can be found in Appendix A.

**Baselines**. We include classic machine learning models such as IForest (Liu et al., 2008), LOF (Breunig et al., 2000), OCSVM (Schölkopf et al., 1999), ECOD (Li et al., 2022), KNN (Cover & Hart, 1967), and PCA (Shyu et al., 2003), which are still widely used and remain strong baselines. Furthermore, 7 deep learning models are considered: DeepSVDD (Ruff et al., 2018), GOAD (Bergman

Table 1: Tabular anomaly detection results in terms of AUC-PR on 20 datasets compared with baseline models. The rank indicates the relative AUC-PR performance within each dataset. The best results are shown in **bold** and the second best in underlined. Standard deviations are provided in Appendix B.

| Dataset | Machine Learning | | | | | | Deep Learning | | | | | | | | |
| | IForest | LOF | OCSVM | ECOD | KNN | PCA | DeepSVDD | GOAD | NeuTraL | ICL | MCM | DRL | Disent | NPTAD | **Ours** |
|---|---|---|---|---|---|---|---|---|---|---|---|---|---|---|---|
| arrhythmia | **0.6412** | 0.5994 | 0.6141 | 0.6212 | 0.6131 | 0.6127 | 0.5711 | 0.4781 | 0.3955 | 0.4657 | 0.5945 | 0.5401 | 0.5953 | 0.3779 | 0.6116 |
| breastw | **0.9949** | 0.9209 | 0.9900 | 0.9922 | 0.9876 | 0.9905 | 0.9841 | 0.8363 | 0.6237 | 0.8376 | 0.9910 | 0.9779 | 0.9802 | 0.9781 | 0.9845 |
| campaign | 0.4570 | 0.4025 | 0.4968 | 0.5010 | 0.4972 | 0.4905 | 0.4453 | 0.1662 | 0.4034 | 0.4877 | 0.4986 | 0.4177 | 0.4196 | 0.4371 | **0.5114** |
| cardio | 0.7941 | 0.6966 | 0.8463 | 0.7144 | 0.7784 | **0.8548** | 0.7939 | 0.3440 | 0.3229 | 0.5748 | 0.8076 | 0.7379 | 0.8507 | 0.8374 | 0.8445 |
| cardiotocography | 0.6943 | 0.6744 | **0.7292** | 0.6539 | 0.6580 | 0.4714 | 0.6987 | 0.4161 | 0.4283 | 0.5723 | 0.6344 | 0.6086 | 0.6856 | 0.7230 | 0.6819 |
| census | 0.1420 | 0.1425 | 0.2059 | 0.1546 | 0.2126 | 0.1168 | 0.1950 | 0.1551 | 0.1705 | 0.1701 | 0.2007 | 0.1826 | 0.1514 | 0.2119 | **0.2496** |
| fraud | 0.2151 | 0.0118 | 0.3502 | 0.3244 | 0.3514 | 0.2627 | 0.2063 | 0.4692 | 0.5902 | **0.7515** | 0.4944 | 0.3510 | 0.2522 | 0.7273 |
| glass | 0.1638 | 0.2458 | 0.1878 | 0.1943 | 0.2454 | 0.1547 | 0.1466 | 0.2791 | **0.5636** | 0.3674 | 0.2294 | 0.2701 | 0.1277 | 0.2964 | 0.2925 |
| ionosphere | 0.9143 | 0.9632 | 0.9750 | 0.7787 | **0.9810** | 0.9173 | 0.8828 | 0.9594 | 0.9411 | 0.9441 | 0.9729 | 0.9704 | 0.9620 | 0.8907 | 0.9785 |
| mammography | 0.3606 | 0.3310 | 0.4016 | **0.5461** | 0.4073 | 0.4279 | 0.4037 | 0.2622 | 0.0678 | 0.1421 | 0.4204 | 0.5331 | 0.4149 | 0.4007 | 0.4154 |
| nslkdd | 0.7730 | 0.9747 | 0.7529 | 0.4941 | 0.9700 | 0.6832 | 0.7499 | 0.8428 | 0.9493 | 0.8852 | **0.9792** | 0.9631 | 0.8466 | 0.7777 | 0.9744 |
| optdigits | 0.1849 | **0.4977** | 0.0675 | 0.0655 | 0.3233 | 0.0559 | 0.0671 | 0.1551 | 0.4672 | 0.3969 | 0.3372 | 0.2727 | 0.1417 | 0.0561 | 0.2170 |
| pendigits | 0.5235 | 0.8508 | 0.5094 | 0.4022 | **0.9681** | 0.3834 | 0.3899 | 0.0281 | 0.5584 | 0.5006 | 0.8381 | 0.6094 | 0.7697 | 0.7044 | 0.8696 |
| pima | **0.7179** | 0.6720 | 0.6915 | 0.6242 | 0.7155 | 0.6930 | 0.6623 | 0.4932 | 0.5619 | 0.6488 | 0.6250 | 0.6759 | 0.6847 | 0.6990 |
| satellite | 0.8428 | 0.8785 | 0.8236 | 0.6614 | 0.8917 | 0.7692 | 0.7433 | 0.8007 | 0.7399 | 0.8636 | 0.8608 | **0.9009** | 0.7932 | 0.8782 | 0.8661 |
| satimage-2 | 0.9261 | 0.9211 | 0.9723 | 0.7446 | 0.9790 | 0.9017 | 0.7816 | 0.9758 | 0.0806 | 0.8861 | 0.8652 | 0.9412 | 0.9658 | **0.9811** | 0.9748 |
| shuttle | 0.9857 | 0.9941 | 0.9732 | 0.9484 | 0.9740 | 0.9613 | 0.9497 | 0.9574 | **0.9981** | 0.9858 | 0.9798 | 0.9893 | 0.9703 | 0.9752 | 0.9883 |
| thyroid | **0.8201** | 0.6781 | 0.7834 | 0.6281 | 0.8109 | 0.7919 | 0.7497 | 0.6937 | 0.1079 | 0.2926 | 0.6930 | 0.6457 | 0.7984 | 0.7604 | 0.7547 |
| wbc | 0.7729 | 0.7800 | 0.7798 | 0.5922 | 0.7661 | 0.7688 | 0.7398 | 0.2629 | 0.0981 | 0.3314 | 0.5548 | 0.7423 | 0.7566 | 0.7395 | **0.7887** |
| wine | 0.5417 | 0.7508 | 0.7561 | 0.3160 | 0.7804 | 0.6385 | 0.5863 | 0.5308 | **0.8331** | 0.5937 | 0.7852 | 0.7486 | 0.8264 | 0.7588 | 0.8254 |
| Average AUC-PR | 0.6233 | 0.6493 | 0.6453 | 0.5479 | 0.6956 | 0.5973 | 0.5873 | 0.5053 | 0.4980 | 0.5768 | 0.6810 | 0.6589 | 0.6541 | 0.6370 | **0.7128** |
| Average Rank | 7.6667 | 7.619 | 6.5714 | 10.5714 | 4.5714 | 9.1429 | 10.5238 | 11.2857 | 10.1905 | 9.4762 | 6.1905 | 7.2857 | 7.5238 | 7.6667 | **3.7143** |

Table 2: Results on synthetic anomaly benchmarks categorized by four anomaly types (Global, Cluster, Local, and Dependency). The upper part reports AUC-PR, while the lower part reports rank across methods. The best results are shown in **bold** and the second best in underlined. Detailed results are provided in Appendix C.

| Synthetic type | IForest | LOF | OCSVM | KNN | PCA | MCM | DRL | Disent | **Ours** |
|---|---|---|---|---|---|---|---|---|---|
| Global | 0.9852 | 0.9616 | 0.9959 | 0.9948 | 0.7671 | 0.9124 | 0.9948 | 0.9705 | **0.9962** |
| Cluster | 0.9221 | 0.9864 | 0.9891 | 0.9846 | 0.7823 | 0.9420 | **0.9934** | 0.9748 | 0.9927 |
| Local | 0.6005 | **0.8101** | 0.6747 | 0.7205 | 0.4977 | 0.6309 | 0.7392 | 0.6775 | 0.7989 |
| Dependency | 0.3968 | 0.7510 | 0.4577 | 0.6345 | 0.2928 | 0.5689 | 0.7992 | 0.3185 | **0.8216** |
| Global (rank) | **2.8182** | 6.4545 | 3.8182 | 3.7273 | 7.6364 | 6.1818 | 3.9091 | 7.0909 | 3.3636 |
| Cluster (rank) | 5.1364 | 4.8636 | 4.1364 | 4.8636 | 6.5455 | 6.2273 | **3.5909** | 5.7727 | 3.8636 |
| Local (rank) | 7.8182 | 2.2727 | 5.1818 | 4.1818 | 8.0000 | 7.0909 | 3.3636 | 5.5455 | **1.5455** |
| Dependency (rank) | 6.8182 | 2.6364 | 6.2727 | 4.5455 | 8.5455 | 4.9091 | 2.0000 | 7.8182 | **1.4545** |

& Hoshen, 2020), NeuTraL (Qiu et al., 2021), ICL (Shenkar & Wolf, 2022), MCM (Yin et al., 2024), DRL (Ye et al., 2025a), and Disent (Ye et al., 2025b). Implementation details of KNN, OCSVM, IForest, LOF, PCA, ECOD, and DeepSVDD are based on PYOD (Zhao et al., 2019), while ICL, NeuTraL, and GOAD are sourced from DeepOD (Xu et al., 2023; 2024). The remaining models are implemented using their official open-source code releases.

## 4.1 MAIN RESULTS

**Real-world Datasets.** As shown in Table 1, LATTE achieves state-of-the-art performance with the highest average AUC-PR of 0.7128. In particular, it surpasses the second-best deep learning model MCM (Yin et al., 2024) by 4.61% in average AUC-PR and by 2.47 in average rank. These results demonstrate the advantage of the proposed attention-based bottleneck and learnable memory bank over prior MLP-based approaches for TAD. Furthermore, compared to NPT-AD, which requires the entire training set as an input, LATTE efficiently captures inter-sample dependencies via the learnable memory bank, consistently outperforming NPT-AD. For detailed runtime analysis, see Appendix E

**Synthetic Anomalies.** Although anomalies can exist in various forms, ADBench (Han et al., 2022) categorizes them into four distinct types. Following this setup, we conduct experiments on synthetic anomalies generated across 10 benchmark datasets and evaluate the performance of LATTE on each anomaly type. Details of the experimental setup for each anomaly type are provided in Appendix C. Global anomalies are samples far from the normal data distribution in the entire feature space. Local anomalies deviate from their local neighborhoods despite appearing globally plausible. Clustered

anomalies are represented as groups of samples that are close to each other but significantly depart from the normal distribution. Dependency anomalies do not follow the typical inter-column correlations observed in normal data. For the details, please refer to Han et al. (2022). As shown in Table 2, most models perform well on Cluster and Global anomalies, whereas baseline methods struggle on Local and Dependency anomalies. LATTE consistently achieves the first or second best model for all anomaly types and outperforms MLP-based reconstruction models. MLP-based reconstruction models such as MCM and Disent show limited performance on Dependency anomalies, as their architectures struggle to capture complex feature interactions, whereas LATTE achieves the best performance by effectively modeling these correlations.

## 4.2 COMPONENT CONTRIBUTION

In this section, we describe how each individual component contributes to the overall performance and clarify its role. In Table 3, all performance gains are reported with respect to a baseline model (1st row) that uses MLP-based encoder and decoder without the memory bank. Incorporating attention mechanisms into encoder and decoder architectures (2nd and 3rd rows) improves the detection performance by 7.9% and 4.4%, respectively. This indicates that attention structures are more suitable for modeling inter column dependencies and heterogeneous attributes than MLPs. Adding the learnable memory module (4th row), which utilizes normal prototypes as described in Section 3.2, provides 1.9% improvements. This supports the choice of the core components in the pro-

Table 3: Ablation study of individual components with standard deviation. In Attn-Enc. and Attn-Dec., ✗ indicates substitution with MLPs, while in Memory it denotes the exclusion of the module.

| Attn-Enc. | Attn-Dec. | Memory | AUC-PR |
|:---:|:---:|:---:|:---:|
| ✗ | ✗ | ✗ | 0.6184 |
| ✓ | ✗ | ✗ | 0.6672 |
| ✗ | ✓ | ✗ | 0.6458 |
| ✗ | ✗ | ✓ | 0.6304 |
| ✓ | ✓ | ✗ | 0.7083 |
| ✓ | ✗ | ✓ | 0.6726 |
| ✗ | ✓ | ✓ | 0.6937 |
| ✓ | ✓ | ✓ | **0.7128** |

posed LATTE architecture. In addition, combining these modules consistently yields further gains. All two-components configurations (5-7th rows) improve their single-component counterparts(2-4th rows), indicating that each module plays a complementary role. Combining all these components (8th row), the model achieves 15.2% relative improvement in AUC-PR compared to the baseline(1st row).

To further explore the effectiveness of introducing attention in TAD, we conduct an additional component analysis with four synthetic anomaly types. The use of MLP yield moderate performance for global (Glo) and cluster (Clu) anomaly types, while MLPs fail to detect anomalies of local (Loc) and dependency (Dep) types as shown in Table 4. In contrast, LATTE, applying attention to

Table 4: Comparison of component variants on synthetic anomaly benchmarks. Glo, Clu, Loc, and Dep represent global, cluster, local, and dependency anomaly type, respectively.

| Attn-Enc. | Attn-Dec. | Glo | Clu | Loc | Dep |
|:---:|:---:|:---:|:---:|:---:|:---:|
| ✗ | ✗ | 0.9615 | 0.9674 | 0.6220 | 0.3005 |
| ✗ | ✓ | 0.9922 | 0.9868 | 0.7023 | 0.6278 |
| ✓ | ✗ | 0.9949 | 0.9853 | 0.7317 | 0.7251 |
| ✓ | ✓ | **0.9962** | **0.9927** | **0.9789** | **0.8216** |

both encoder and decoder achieves significantly better performance across all anomaly types. This highlights that introducing attention in TAD helps the model to capture intrinsic inter-column dependencies and detect anomalies that violate these patterns. These results confirm that integrating attention is crucial for effectively handling complex anomalies that standard MLP-based methods struggle to identify.

## 4.3 DESIGN CHOICES AND ABLATIONS

In this section, we conduct ablation studies on the core components of our architecture: the encoder with a latent bottleneck, the learnable memory bank, and the query-based decoder. Specifically, we analyze how encoder performance varies with latent capacity and layer depth, how memory behavior depends on capacity and temperature, and how different query formulations affect decoding, thereby clarifying the operating conditions under which each component is most effective.

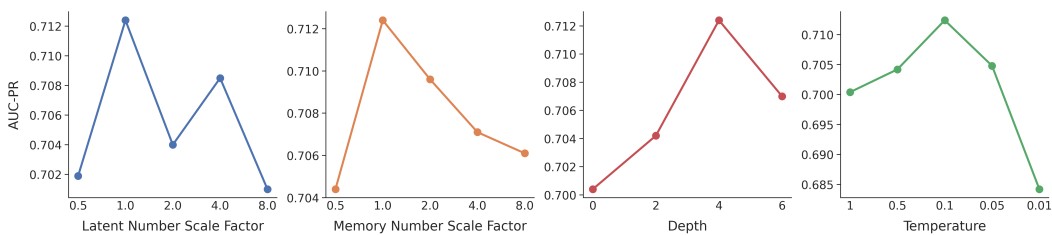

Figure 2: Hyperparameter sensitivity analysis

**Latent Bottleneck and Memory Bank.** We conduct how different configurations of the latent bottleneck and memory bank affect performance. First, regarding size, the latent bottleneck and memory bank respectively encode information from the columns and the training samples, thereby compactly capturing the heterogeneous patterns of normal data. Accordingly, we set the size of the latent bottleneck to be approximately $\sqrt{F}$ and the size of the memory bank to $\sqrt{N_{\text{train}}}$. As shown in Figure 2, we vary the number of latent tokens and memory size by scaling their default values with $s \in \{0.5, 1.0, 2.0, 4.0, 8.0\}$. The results show that while the default setting ($s = 1.0$) achieves the best performance, insufficient sizes limit the model's ability to represent normal patterns, whereas excessively large sizes result in nearly identity mapping rather than learning meaningful correlations for reconstruction. We further investigate the effect of latent self-attention depth $L$. The results show that performance improves as $L$ increases up to 4, after which it begins to decline, indicating that $L = 4$ provides the optimal depth. This suggests that shallow layers lack the capacity to capture complex inter-column dependencies, whereas excessive depth leads to diminished generalization. Lastly, temperature $\tau$ in latent memory bank controls whether attention weights are distributed more uniformly or more sparsely across memory slots. While extreme values such as $1.0$ and $0.01$ result in overly uniform or excessively sparse distributions that degrade performance, we find that $0.1$ provides the optimal balance and achieves the best result.

**Query Design.** We investigate different query formulations by comparing global-only ($\mathbf{q}_{\texttt{global}}$), local-only ($\mathbf{b}_i$), and the combined global-local design ($\mathbf{q}_{\texttt{global}} + \mathbf{b}_i$). As shown in Table 5, incorporating both global and local components improves performance compared to using either alone. Specifically, the combined design yields 2.2% and 0.5% gains over local-only and global-only queries, respectively, validating the effectiveness of our proposed query formulation.

Table 5: Ablation study on query composition.

| Global | Local | AUC-PR |
|:---:|:---:|:---:|
| ✗ | ✓ | 0.6972 |
| ✓ | ✗ | 0.7089 |
| ✓ | ✓ | **0.7124** |

### 4.4 ANALYSIS VIA VISUALIZATION

**T-SNE Analysis.** As shown in Figure 3, we visualize the original and reconstructed representations of both normal and local anomaly samples via T-SNE (van der Maaten & Hinton, 2008). Notably, with the construction of our memory bank, abnormal samples that were originally located away from the normal cluster are reconstructed to align more closely with the distribution of normal samples. This indicates that anomaly samples are not simply reconstructed in an arbitrary manner but are mapped toward the manifold of normal samples represented by memory vectors. For additional analysis with visualizations, see Appendix E.

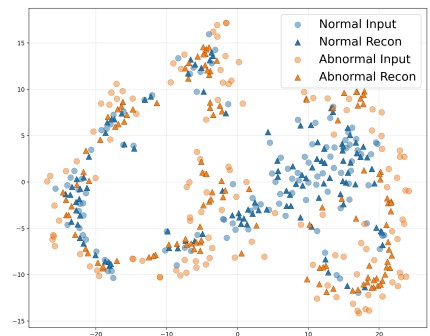

Figure 3: T-SNE visualization.

**Normal vs. Abnormal Analysis.** We further investigate the behavior of attention maps to understand how LATTE distinguishes between normal and abnormal samples. For this, we consider two attention maps: (*i*) decoder attention map created by latent before memory addressing, (*ii*) decoder attention map generated by latent after memory addressing is applied. As shown in Figure 7, normal sample's high-scored attention position is same in (*i*) and (*ii*). However, on abnormal sample, after the memory addressing is applied, the decoder attention

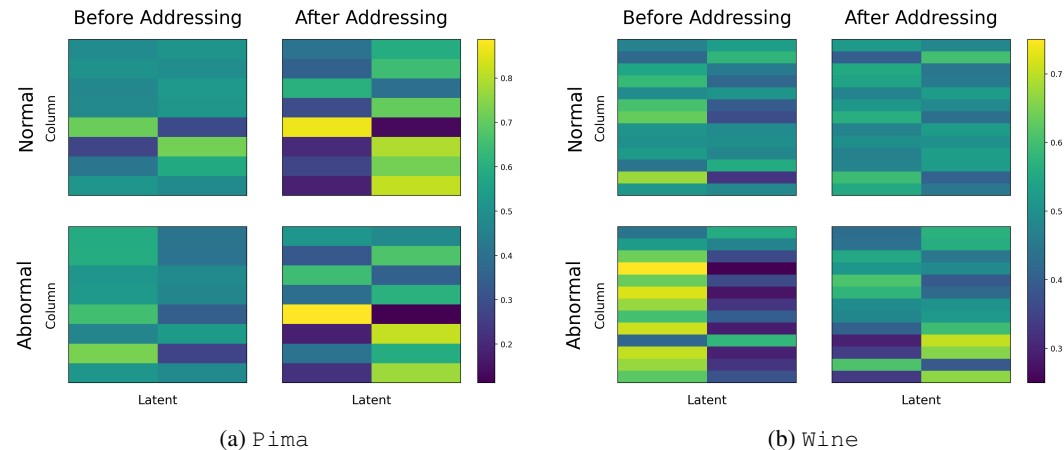

(a) `Pima`    (b) `Wine`

Figure 4: Visualization of attention maps on the datasets. The first row corresponds to the average attention map of normal samples, while the second row corresponds to an abnormal sample. Columns represent different attention types: latent-to-query cross-attention without memory addressing and latent-to-query cross-attention with memory addressing.

map becomes closer to that of normal samples. Also, in the second case, in the decoder attention map, on average, the normal samples have similar decoder attention maps before and after memory addressing. However, the abnormal sample has largely different decoder attention map after memory addressing compared to that of before memory addressing, which means that unseen decoder attention map cannot be represented after memory addressing. Above two results highlight that the model is trained to use only observed main patterns during training for reconstruction.

## 5 RELATED WORK

**Classical AD methods**. In unsupervised anomaly detection, classical methods remain strong baselines, especially on tabular data. Broadly, density-based approaches such as KDE (Hastie et al., 2009) and GMM estimate the normal data distribution, while ECOD (Li et al., 2022) leverages empirical cumulative distributions. Neighbor methods includes KNN (Cover & Hart, 1967) and LOF (Breunig et al., 2000). IForest (Liu et al., 2008) identifies anomalies as instances that are isolated with fewer of splits. OCSVM (Schölkopf et al., 1999) learn a decision boundary that encloses normal data in a kernel feature space. PCA (Shlens, 2014) reconstructs the input with fewer principal components.

**Deep Learning-based Approaches**. Deep learning-based TAD methods employ diverse strategies to model normal patterns. Reconstruction-based approaches like MCM (Yin et al., 2024) utilize *masked modeling*, forcing the network to recover missing information from partial inputs, and NPT-AD and its variant (Thimonier et al., 2024a;b) leverage attention between samples to reconstruct masked and augmented version of input, respectively. In contrast, GOAD (Bergman & Hoshen, 2020) and NeuTraL (Golan & El-Yaniv, 2018) focus on *transformation-based representation learning*, whereas DeepSVDD (Ruff et al., 2018) and ICL (Shenkar & Wolf, 2022) leverage *contrastive learning approach* to constrain normal samples within a compact embedding space. Regarding latent structure, DRL (Ye et al., 2025a) and Disent (Ye et al., 2025b) aim to *disentangle* the feature space into basis vectors or separate subspaces. Distinct from these approaches, LATTE does not rely on input perturbations (masking or transformations) or rigid constraints on latent space. Instead, LATTE employs an attention-based bottleneck to directly capture intrinsic inter-column dependencies within a pure reconstruction framework, complemented by a learnable memory module that retrieves prototypical normal patterns to guide the reconstruction process.

**Deep Learning for Tabular Data**. Many deep learning models have been proposed for tabular learning. TabTransformer (Huang et al., 2020) applies a Transformer to categorical features, whereas FT-Transformer (Gorishniy et al., 2021) tokenizes all columns and applies a Transformer to handle heterogeneous features. SAINT (Song et al., 2019) augments supervised training with masked mod-

eling as self-supervision and performs attention across columns and samples. TabPFN (Hollmann et al., 2023) pretrains a large Transformer on synthetic tasks to leverage in-context learning. More recently, TabR (Gorishniy et al., 2024) and MNCA (Ye et al., 2024) proposed retrieval-based approaches inspired by traditional methods, reporting competitive performance. Building upon this momentum, we introduce LATTE to establish that bridging deep and traditional methods yields state-of-the-art performance in TAD. For comprehensive survey on tabular learning, please refer to (Borisov et al., 2024).

## 6 CONCLUSION

In this paper, we propose LATTE, a simple yet effective reconstruction-based framework for tabular anomaly detection (TAD) that leverages an attention-based bottleneck architecture. Specifically, we revisit prior approaches and present a novel design that captures heterogeneous feature characteristics and inter-column relationships more effectively for reconstruction. Through extensive experiments on 20 diverse datasets, we demonstrate that our method achieves state-of-the-art performance. Furthermore, we conduct comprehensive ablation studies to clarify the role of each component and justify our architectural choices. We highlight two key takeaways: (*i*) architectural choices are crucial in TAD yet underexplored; and (*ii*) incorporating core principles of traditional methods into modern deep architecture improves detection performance. We believe that this study motivates future research to prioritize architectural designs that bridge the gap between deep architectures and traditional machine learning models in TAD.

**Ethics Statement.** In this paper, we revisit architectural choices for tabular anomaly detection and do not foresee any ethical concerns.

**Reproducibility Statement.** We provide code that contains our experimental setup and model architecture in the supplementary materials, implemented in PyTorch (Paszke et al., 2019). All experiments are conducted on NVIDIA RTX 4090 GPUs.

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

## A  HYPERPARAMETERS DETAILS

In this section, we describe the implementation details of LATTE used in our experiments. From the optimization perspective, we use the AdamW optimizer (Loshchilov & Hutter, 2019) with a learning rate of 0.001 and weight decay of $1 \times 10^{-5}$. An exponential learning rate scheduler (Li & Arora, 2020) with a decay rate of 0.98 is applied during training. Furthermore, early stopping is employed with a patience of 10 and 20 epochs for small-scale and large-scale datasets, respectively. The batch size is set to 512 for most datasets, while larger datasets such as Census and Fraud are trained with a batch size of 2048. Regarding model hyperparameters, the model dimension $d$ is 64, and the FFNs inside the attention modules expand this to 256 with a ratio of 4. The latent self-attention layers are assigned a depth $L$ of 4, while all cross-attention layers use a depth of 1. In the latent–memory cross-attention, the temperature $\tau$ is set to 0.1. The above settings remain fixed across all experiments, whereas the numbers of latent tokens and memory vectors vary with the dataset. Specifically, the number of latent tokens is chosen as the largest power of $n \in \mathbb{Z}$ smaller than $\sqrt{F}$, where $F$ denotes the number of features, and the number of memory vectors is chosen as the largest power of $n$ smaller than $\sqrt{|\mathcal{D}_{\texttt{train}}|}$, where $|\mathcal{D}_{\texttt{train}}|$ is the size of the training set. We provide the detailed process of selecting these hyperparameters in Section 4.3.

## B  DETAILED EXPERIMENTAL RESULTS

Table 6: Tabular anomaly detection results in terms of AUC-PR with standard deviation on 20 datasets, compared with baseline models. The rank indicates the relative AUC-PR performance within each dataset. The best results are shown in **bold** and the second best in underlined.

| Dataset | Machine Learning | | | | | | Deep Learning | | | | | | | | |
| --- | IForest | LOF | OCSVM | ECOD | KNN | PCA | DeepSVDD | GOAD | NeuTraL | ICL | MCM | DRL | Disent | NPTAD | Ours |
| arrhythmia | **0.6412**±0.0422 | 0.5994±0.0311 | 0.6141±0.0270 | 0.6212±0.0321 | 0.6131±0.0282 | 0.6127±0.0276 | 0.5711±0.0413 | 0.4781±0.0671 | 0.3955±0.0393 | 0.4657±0.0252 | 0.5945±0.0264 | 0.5401±0.0527 | 0.5953±0.0277 | 0.3779±0.0855 | 0.6116±0.0289 |
| breastw | **0.9949**±0.0017 | 0.9209±0.0116 | 0.9900±0.0076 | 0.9922±0.0013 | 0.9876±0.0053 | 0.9905±0.0055 | 0.9841±0.0079 | 0.8363±0.0877 | 0.6237±0.0384 | 0.8376±0.0471 | 0.9910±0.0057 | 0.9779±0.0096 | 0.9802±0.0096 | 0.9781±0.0062 | 0.9845±0.0070 |
| campaign | 0.4370±0.0142 | 0.4025±0.0109 | 0.4968±0.0033 | 0.5910±0.0029 | 0.4972±0.0034 | 0.4905±0.0034 | 0.4453±0.0408 | 0.1662±0.0061 | 0.4034±0.0340 | 0.4877±0.0535 | 0.4986±0.0047 | 0.4177±0.0632 | 0.4196±0.0388 | 0.4371±0.0739 | **0.5514**±0.0084 |
| cardio | 0.7941±0.0216 | 0.6966±0.0281 | 0.8463±0.0170 | 0.7144±0.0103 | 0.7784±0.0206 | **0.8848**±0.0173 | 0.7939±0.0841 | 0.3440±0.0483 | 0.3229±0.0291 | 0.5748±0.0753 | 0.8076±0.0495 | 0.7379±0.0313 | 0.8507±0.0193 | 0.8374±0.0162 | 0.8445±0.0204 |
| cardiotocography | 0.6943±0.0315 | 0.6744±0.0192 | 0.7292±0.0130 | 0.6539±0.0055 | 0.6580±0.0112 | 0.4714±0.1775 | 0.6987±0.0819 | 0.4161±0.0193 | 0.4283±0.0116 | 0.5723±0.0254 | 0.6344±0.0147 | 0.6086±0.0311 | 0.6856±0.0312 | **0.7420**±0.0145 | 0.6819±0.0161 |
| census | 0.1420±0.0073 | 0.1425±0.0011 | 0.2059±0.0027 | 0.1546±0.0002 | 0.2126±0.0013 | 0.1168±0.0000 | 0.1950±0.0104 | 0.1551±0.0162 | 0.1705±0.0172 | 0.1701±0.0074 | 0.2007±0.0048 | 0.1826±0.0231 | 0.1514±0.0345 | 0.2119±0.0052 | **0.2496**±0.0195 |
| fraud | 0.2151±0.0386 | 0.0118±0.0031 | 0.3502±0.0108 | 0.3244±0.0092 | 0.3514±0.0109 | 0.2627±0.0083 | 0.2063±0.0921 | 0.4692±0.0422 | 0.6487±0.0655 | 0.5902±0.0545 | **0.7515**±0.0076 | 0.4994±0.1268 | 0.3510±0.0668 | 0.2522±0.0067 | 0.7223±0.0191 |
| glass | 0.1638±0.0279 | 0.2458±0.0574 | 0.1878±0.0394 | 0.1943±0.0334 | 0.2454±0.0452 | 0.1547±0.0371 | 0.1466±0.0400 | 0.2791±0.1281 | **0.5636**±0.1357 | 0.3647±0.0847 | 0.2294±0.0500 | 0.2701±0.0584 | 0.1277±0.0241 | 0.2964±0.0771 | 0.2925±0.0518 |
| ionosphere | 0.9143±0.0142 | 0.9632±0.0107 | 0.9750±0.0059 | 0.7787±0.0096 | **0.9810**±0.0054 | 0.9173±0.0123 | 0.8828±0.0282 | 0.9594±0.0129 | 0.9411±0.0127 | 0.9441±0.0232 | 0.9729±0.0053 | 0.9704±0.0112 | 0.9620±0.0143 | 0.8907±0.0700 | 0.9785±0.0055 |
| mammography | 0.3606±0.0437 | 0.3310±0.0213 | 0.4016±0.0171 | **0.5461**±0.0092 | 0.4073±0.0246 | 0.4279±0.0248 | 0.4037±0.0831 | 0.2622±0.0301 | 0.0678±0.0041 | 0.1421±0.0548 | 0.4204±0.0992 | 0.5331±0.0275 | 0.4149±0.0509 | 0.4007±0.0306 | 0.4154±0.0224 |
| nslkdd | 0.7730±0.0288 | 0.9747±0.0026 | 0.7529±0.0025 | 0.4941±0.0002 | 0.9700±0.0026 | 0.6832±0.0000 | 0.7499±0.0320 | 0.8428±0.0431 | 0.9493±0.0172 | 0.8852±0.0784 | **0.9792**±0.0025 | 0.9631±0.0589 | 0.8466±0.0718 | 0.7777±0.0231 | 0.9744±0.0058 |
| optdigits | 0.1849±0.0725 | **0.4977**±0.0272 | 0.0675±0.0015 | 0.0655±0.0009 | 0.3233±0.0167 | 0.0589±0.0000 | 0.0671±0.0230 | 0.1551±0.0350 | 0.4672±0.0890 | 0.3969±0.0797 | 0.3372±0.0378 | 0.2727±0.1478 | 0.1417±0.0460 | 0.0561±0.0509 | 0.2170±0.0360 |
| pendigits | 0.5235±0.0457 | 0.3858±0.0259 | 0.5094±0.0286 | 0.4022±0.0184 | **0.9681**±0.0150 | 0.3834±0.0162 | 0.3899±0.2159 | 0.0281±0.0015 | 0.5584±0.0415 | 0.5006±0.1259 | 0.3381±0.1075 | 0.6094±0.2143 | 0.7697±0.0589 | 0.7044±0.1175 | 0.8696±0.0414 |
| pima | **0.7179**±0.0202 | 0.6720±0.0221 | 0.6915±0.0232 | 0.6242±0.0165 | 0.7155±0.0236 | 0.6930±0.0236 | 0.6623±0.0549 | 0.4932±0.0228 | 0.5619±0.0219 | 0.6488±0.0213 | 0.6250±0.0795 | 0.6322±0.0411 | 0.6759±0.0282 | 0.6847±0.0213 | 0.6990±0.0345 |
| satellite | 0.8428±0.0071 | 0.8785±0.0040 | 0.8236±0.0017 | 0.6614±0.0041 | 0.8917±0.0033 | 0.7692±0.0018 | 0.7433±0.0477 | 0.8007±0.0049 | 0.7399±0.0082 | 0.8636±0.0103 | 0.8608±0.0070 | **0.9009**±0.0126 | 0.7932±0.0164 | 0.8782±0.0244 | 0.8661±0.0036 |
| satimage-2 | 0.9261±0.0189 | 0.9221±0.0208 | 0.9723±0.0015 | 0.7446±0.0121 | 0.9790±0.0103 | 0.9017±0.0027 | 0.7816±0.2320 | 0.4937±0.0228 | 0.8861±0.0390 | 0.8861±0.0390 | 0.8652±0.0564 | 0.9412±0.0197 | 0.9658±0.0220 | **0.9811**±0.0031 | 0.9748±0.0038 |
| shuttle | 0.9857±0.0047 | 0.9941±0.0022 | 0.9732±0.0023 | 0.9484±0.0029 | 0.9740±0.0043 | 0.9613±0.0040 | 0.9497±0.0262 | 0.9574±0.0039 | **0.9981**±0.0009 | 0.9883±0.0086 | 0.9798±0.0046 | 0.9893±0.0057 | 0.9703±0.0105 | 0.9752±0.0088 | 0.9883±0.0058 |
| thyroid | **0.8201**±0.0375 | 0.6781±0.0633 | 0.7834±0.0238 | 0.6281±0.0257 | 0.8109±0.0208 | 0.7919±0.0156 | 0.7497±0.1018 | 0.6937±0.0178 | 0.1079±0.0103 | 0.2926±0.1020 | 0.6930±0.0602 | 0.6457±0.0632 | 0.7984±0.0245 | 0.7604±0.0567 | 0.7547±0.0330 |
| wbc | 0.7729±0.0444 | 0.7800±0.0246 | 0.7798±0.0302 | 0.5922±0.0471 | 0.7661±0.0330 | 0.7688±0.0323 | 0.7398±0.1163 | 0.2629±0.0333 | 0.0981±0.0093 | 0.3314±0.1088 | 0.5548±0.1872 | 0.7423±0.0488 | 0.7566±0.0399 | 0.7395±0.0688 | **0.7887**±0.0305 |
| wine | 0.5417±0.1018 | 0.7508±0.0895 | 0.7561±0.0655 | 0.3160±0.0391 | **0.7804**±0.0787 | 0.6385±0.0834 | 0.5863±0.2659 | 0.5308±0.1011 | **0.8331**±0.1455 | 0.5937±0.0244 | 0.7852±0.1410 | 0.7486±0.1532 | 0.8264±0.0002 | 0.7588±0.0813 | 0.8254±0.1193 |
| Average AUC-PR | 0.6233±0.0312 | 0.6493±0.0239 | 0.6453±0.0162 | 0.5479±0.0141 | 0.6956±0.0179 | 0.5973±0.0247 | 0.5873±0.0847 | 0.5053±0.0361 | 0.4980±0.0373 | 0.5766±0.0623 | 0.6810±0.0479 | 0.6589±0.0581 | 0.6541±0.0338 | 0.6370±0.0398 | **0.7128**±0.0256 |
| Average Rank | 7.6667 | 7.619 | 6.5714 | 10.5714 | 4.5714 | 9.1429 | 10.5238 | 11.2857 | 10.1905 | 9.4762 | 6.1905 | 7.2857 | 7.5238 | 7.6667 | **3.7143** |

Table 7: Tabular anomaly detection results in terms of F1 with standard deviation on 20 datasets, compared with baseline models. The rank indicates the relative F1 performance within each dataset. The best results are shown in **bold** and the second best in underlined.

| Dataset | Machine Learning | | | | | | Deep Learning | | | | | | | | |
| --- | IForest | LOF | OCSVM | ECOD | KNN | PCA | DeepSVDD | GOAD | NeuTraL | ICL | MCM | DRL | Disent | NPTAD | Ours |
| arrhythmia | **0.6045**±0.0208 | 0.5530±0.0205 | 0.5621±0.0281 | 0.5597±0.0199 | 0.5667±0.0269 | 0.5636±0.0284 | 0.5606±0.0385 | 0.4409±0.0873 | 0.4045±0.0392 | 0.4534±0.0265 | 0.5530±0.0270 | 0.5121±0.0557 | 0.5455±0.0319 | 0.3212±0.0909 | 0.5591±0.0307 |
| breastw | **0.9724**±0.0040 | 0.9218±0.0125 | 0.9661±0.0050 | 0.9540±0.0056 | 0.9657±0.0068 | 0.9644±0.0057 | 0.9596±0.0063 | 0.7144±0.1208 | 0.6740±0.0530 | 0.8297±0.0302 | 0.9669±0.0057 | 0.9439±0.0086 | 0.9431±0.0277 | 0.9439±0.0136 | 0.9540±0.0076 |
| campaign | 0.4341±0.0109 | 0.4223±0.0109 | 0.4882±0.0015 | 0.6648±0.0017 | 0.5060±0.0025 | 0.4914±0.0019 | 0.4593±0.0270 | 0.1442±0.0115 | 0.4517±0.0301 | 0.5057±0.0614 | **0.5300**±0.0050 | 0.4577±0.0643 | 0.4428±0.0136 | 0.4466±0.0573 | 0.5308±0.0076 |
| cardio | 0.7114±0.0329 | 0.6506±0.0307 | 0.7812±0.0297 | 0.6648±0.0147 | 0.6920±0.0273 | **0.7994**±0.0241 | 0.7239±0.0171 | 0.2778±0.0594 | 0.5244±0.0841 | 0.7500±0.0515 | 0.6574±0.0143 | **0.8028**±0.0383 | 0.7898±0.0339 | 0.7472±0.0319 | |
| cardiotocography | 0.6315±0.0309 | 0.6264±0.0121 | 0.6376±0.0079 | 0.6258±0.0038 | 0.5828±0.0091 | 0.1908±0.3072 | 0.6195±0.0924 | 0.3221±0.0158 | 0.4341±0.0169 | 0.5107±0.0263 | 0.5451±0.0250 | 0.5049±0.0448 | 0.5940±0.0250 | **0.6451**±0.0114 | 0.5888±0.0178 |
| census | 0.1069±0.0118 | 0.1353±0.0024 | 0.2129±0.0025 | 0.1222±0.0005 | 0.2216±0.0025 | 0.0000±0.0000 | 0.2008±0.0149 | 0.1799±0.0241 | 0.1917±0.0228 | 0.1655±0.0094 | 0.1916±0.0103 | 0.2054±0.0318 | 0.1491±0.0444 | 0.2263±0.0069 | **0.2857**±0.0211 |
| fraud | 0.3108±0.0349 | 0.0000±0.0000 | 0.4543±0.0109 | 0.3817±0.0049 | 0.4858±0.0127 | 0.3307±0.0072 | 0.2868±0.0764 | 0.5915±0.0482 | 0.7117±0.0693 | 0.6244±0.0470 | **0.8008**±0.0000 | 0.5352±0.1135 | 0.3783±0.0488 | 0.3246±0.0169 | 0.7514±0.0171 |
| glass | 0.1223±0.0631 | 0.2090±0.0468 | 0.1111±0.0000 | 0.1667±0.0586 | 0.1333±0.0468 | 0.1111±0.0000 | 0.1000±0.0631 | 0.2667±0.1828 | **0.4889**±0.1673 | 0.2889±0.1304 | 0.2000±0.0703 | 0.2000±0.0676 | 0.1111±0.0000 | 0.2222±0.0987 | 0.1889±0.0705 |
| ionosphere | 0.8095±0.0221 | 0.8873±0.0336 | 0.9214±0.0223 | 0.6611±0.0122 | 0.9278±0.0145 | 0.8056±0.0272 | 0.7778±0.0446 | 0.8559±0.0338 | 0.8738±0.0127 | 0.8706±0.0191 | 0.8968±0.0099 | 0.9024±0.0202 | 0.8251±0.0969 | 0.7960±0.0609 | 0.9079±0.0228 |
| mammography | 0.3742±0.0374 | 0.3815±0.0101 | 0.4173±0.0108 | **0.5335**±0.0036 | 0.4050±0.0136 | 0.4700±0.0087 | 0.4369±0.0993 | 0.3231±0.0254 | 0.1000±0.0132 | 0.1856±0.0619 | 0.4427±0.0893 | 0.5281±0.0184 | 0.4292±0.0428 | 0.4704±0.0231 | 0.4135±0.0273 |
| nslkdd | 0.8320±0.0077 | 0.9414±0.0045 | 0.7340±0.0036 | 0.4900±0.0007 | 0.9640±0.0107 | 0.0000±0.0000 | 0.7221±0.0449 | 0.8024±0.0433 | 0.8911±0.0224 | 0.8069±0.0116 | 0.9599±0.0025 | 0.9244±0.0293 | 0.8251±0.0669 | 0.7383±0.0326 | 0.9537±0.0086 |
| optdigits | 0.1900±0.1212 | **0.5947**±0.0402 | 0.0040±0.0034 | 0.0267±0.0000 | 0.3088±0.0440 | 0.0000±0.0000 | 0.0187±0.0224 | 0.3834±0.0615 | 0.5113±0.0868 | 0.4540±0.0939 | 0.3642±0.0506 | 0.2667±0.2084 | 0.0827±0.0884 | 0.0047±0.0071 | 0.1540±0.0839 |
| pendigits | 0.5301±0.0349 | 0.8045±0.0257 | 0.5321±0.0179 | 0.4282±0.0173 | **0.9340**±0.0148 | 0.4436±0.0150 | 0.4154±0.2082 | 0.0051±0.0079 | 0.5859±0.0357 | 0.4872±0.1145 | 0.6175±0.0161 | 0.6138±0.0743 | 0.6369±0.0260 | 0.7055±0.0508 | 0.8186±0.0054 |
| pima | 0.6888±0.0120 | 0.6672±0.0149 | 0.6843±0.0077 | 0.5784±0.0077 | **0.6989**±0.0109 | 0.6925±0.0129 | 0.6466±0.0514 | 0.6707±0.0947 | 0.6900±0.0043 | 0.5623±0.0189 | 0.7367±0.0065 | 0.6369±0.0260 | 0.6750±0.0265 | 0.6578±0.0135 | 0.6750±0.0189 |
| satellite | 0.6899±0.0069 | 0.7545±0.0053 | 0.6831±0.0006 | 0.5398±0.0016 | 0.7646±0.0030 | 0.6133±0.0026 | 0.6064±0.0403 | 0.6707±0.0947 | 0.6900±0.0043 | 0.7522±0.0140 | 0.7367±0.0065 | **0.8052**±0.0173 | 0.6549±0.0427 | 0.7766±0.0295 | 0.7454±0.0001 |
| satimage-2 | 0.8704±0.0310 | 0.8592±0.0122 | 0.9324±0.0089 | 0.7197±0.0045 | 0.9352±0.0165 | 0.8479±0.0111 | 0.7534±0.2064 | 0.6707±0.0947 | 0.6900±0.0043 | 0.8056±0.0530 | 0.7789±0.0510 | 0.8859±0.0289 | 0.9408±0.0303 | **0.9521**±0.0119 | 0.9254±0.0116 |
| shuttle | 0.9654±0.0066 | 0.9822±0.0015 | 0.9674±0.0065 | 0.9163±0.0020 | 0.9815±0.0015 | 0.9593±0.0039 | 0.9392±0.0630 | 0.9737±0.0054 | **0.9868**±0.0015 | 0.9725±0.0170 | 0.9860±0.0017 | 0.9847±0.0027 | 0.9722±0.0023 | 0.9795±0.0031 | 0.9856±0.0035 |
| thyroid | **0.8032**±0.0304 | 0.6022±0.0651 | 0.7516±0.0235 | 0.6183±0.0170 | 0.7462±0.0321 | 0.7194±0.0275 | 0.6785±0.1159 | 0.7097±0.0190 | 0.0129±0.0208 | 0.3828±0.1044 | 0.6839±0.0306 | 0.6362±0.0546 | 0.7376±0.0177 | 0.6914±0.0606 | 0.6914±0.0359 |
| wbc | 0.6952±0.0333 | 0.6714±0.0270 | 0.5429±0.0533 | 0.6183±0.0070 | 0.6619±0.0151 | 0.6190±0.0094 | 0.6381±0.0704 | 0.2143±0.0337 | 0.1000±0.0333 | 0.3095±0.0985 | 0.4952±0.1756 | 0.6476±0.0400 | 0.6952±0.0246 | 0.6571±0.0492 | **0.7000**±0.0321 |
| wine | 0.6100±0.0876 | 0.7000±0.0616 | 0.7100±0.0738 | 0.3600±0.1075 | 0.7400±0.0843 | 0.6400±0.0843 | 0.5100±0.3107 | 0.4000±0.1054 | **0.7700**±0.0823 | 0.5700±0.1947 | 0.7500±0.1354 | 0.6800±0.1135 | 0.6500±0.0972 | 0.7000±0.1054 | 0.7300±0.0949 |
| Average AUC-PR | 0.5976±0.0322 | 0.6178±0.0233 | 0.6115±0.0146 | 0.5228±0.0162 | 0.6611±0.0200 | 0.5157±0.0302 | 0.5527±0.0475 | 0.4714±0.0462 | 0.4881±0.0386 | 0.5576±0.0658 | 0.6520±0.0462 | 0.6260±0.0596 | 0.6100±0.0365 | 0.6034±0.0412 | **0.6653**±0.0294 |
| Average Rank | 7.881 | 7.7857 | 6.5238 | 10.6667 | 4.5238 | 9.5238 | 10.5238 | 11 | 9.4762 | 9.5238 | 5.8095 | 7.0238 | 7.881 | 7.1905 | 4.6667 |

In this section, we present the main results in Section 4.1 with standard deviation. As shown in Table 6, conventional machine learning methods generally show low variability due to their predominantly non-parametric nature, while deep learning models, given their large parameterization, tend to exhibit higher variance. Nevertheless, our method not only exhibits most robust performance than all deep learning baselines but also is even competitive with conventional machine learning methods. Furthermore, we provide the main results with different metrics such as F1-score and AUROC with standard deviation in Table 7 and 8 for each. We also provide ablation study results with standard deviations in Table 9.

Table 8: Tabular anomaly detection results in terms of AUC-ROC with standard deviation on 20 datasets, compared with baseline models. The rank indicates the relative AUC-ROC performance within each dataset. The best results are shown in **bold** and the second best in underlined.

| | Machine Learning | | | | | | Deep Learning | | | | | | | | |
|---|---|---|---|---|---|---|---|---|---|---|---|---|---|---|---|
| Dataset | IForest | LOF | OCSVM | ECOD | KNN | PCA | DeepSVDD | GOAD | NeuTraL | ICL | MCM | DRL | Disent | NPTAD | Ours |
| arrhythmia | **0.8203**$^{\pm0.0130}$ | 0.8038$^{\pm0.0142}$ | 0.8032$^{\pm0.0123}$ | 0.8048$^{\pm0.0109}$ | 0.8065$^{\pm0.0137}$ | 0.7998$^{\pm0.0130}$ | 0.7635$^{\pm0.0366}$ | 0.6364$^{\pm0.0786}$ | 0.6807$^{\pm0.0289}$ | 0.7503$^{\pm0.0283}$ | 0.7942$^{\pm0.0128}$ | 0.7483$^{\pm0.0051}$ | 0.7852$^{\pm0.0151}$ | 0.5601$^{\pm0.0987}$ | 0.8038$^{\pm0.0151}$ |
| breastw | **0.9951**$^{\pm0.0016}$ | 0.9588$^{\pm0.0082}$ | 0.9916$^{\pm0.0038}$ | 0.9920$^{\pm0.0013}$ | 0.9901$^{\pm0.0031}$ | 0.9917$^{\pm0.0030}$ | 0.9879$^{\pm0.0038}$ | 0.7540$^{\pm0.1285}$ | 0.7189$^{\pm0.0358}$ | 0.8883$^{\pm0.0178}$ | 0.9921$^{\pm0.0033}$ | 0.9819$^{\pm0.0051}$ | 0.9803$^{\pm0.0134}$ | 0.9824$^{\pm0.0046}$ | 0.9869$^{\pm0.0041}$ |
| campaign | 0.7333$^{\pm0.0109}$ | 0.7028$^{\pm0.0074}$ | 0.7761$^{\pm0.0012}$ | 0.7699$^{\pm0.0008}$ | 0.7851$^{\pm0.0022}$ | 0.7699$^{\pm0.0013}$ | 0.7235$^{\pm0.0303}$ | 0.3790$^{\pm0.0098}$ | 0.7120$^{\pm0.0450}$ | **0.8000**$^{\pm0.0455}$ | 0.7825$^{\pm0.0642}$ | 0.7262$^{\pm0.0504}$ | 0.6981$^{\pm0.0140}$ | 0.7239$^{\pm0.0698}$ | 0.7985$^{\pm0.0074}$ |
| cardio | 0.9489$^{\pm0.0097}$ | 0.9265$^{\pm0.0090}$ | 0.9633$^{\pm0.0045}$ | 0.9356$^{\pm0.0025}$ | 0.9308$^{\pm0.0078}$ | **0.9659**$^{\pm0.0042}$ | 0.9386$^{\pm0.0275}$ | 0.5109$^{\pm0.0462}$ | 0.6873$^{\pm0.0273}$ | 0.8135$^{\pm0.0585}$ | 0.9428$^{\pm0.0178}$ | 0.8950$^{\pm0.0267}$ | 0.9570$^{\pm0.0077}$ | 0.9538$^{\pm0.0080}$ | 0.9528$^{\pm0.0082}$ |
| cardiotocography | 0.8080$^{\pm0.0251}$ | 0.7817$^{\pm0.0130}$ | 0.8208$^{\pm0.0070}$ | 0.7843$^{\pm0.0038}$ | 0.7360$^{\pm0.0071}$ | 0.5976$^{\pm0.1571}$ | 0.7867$^{\pm0.0801}$ | 0.4226$^{\pm0.0183}$ | 0.5659$^{\pm0.0091}$ | 0.6593$^{\pm0.0256}$ | 0.7263$^{\pm0.0196}$ | 0.6668$^{\pm0.0381}$ | 0.7622$^{\pm0.0214}$ | **0.8332**$^{\pm0.0123}$ | 0.7418$^{\pm0.0155}$ |
| census | 0.6275$^{\pm0.0167}$ | 0.6037$^{\pm0.0032}$ | 0.7025$^{\pm0.0010}$ | 0.6593$^{\pm0.0005}$ | 0.7198$^{\pm0.0013}$ | 0.5000$^{\pm0.0000}$ | 0.6929$^{\pm0.0114}$ | 0.5983$^{\pm0.0400}$ | 0.6124$^{\pm0.0522}$ | 0.6548$^{\pm0.0132}$ | 0.7016$^{\pm0.0074}$ | 0.6531$^{\pm0.0477}$ | 0.5766$^{\pm0.0757}$ | 0.7089$^{\pm0.0042}$ | **0.7312**$^{\pm0.0198}$ |
| fraud | 0.9495$^{\pm0.0033}$ | 0.7606$^{\pm0.0468}$ | 0.9562$^{\pm0.0001}$ | 0.9480$^{\pm0.0002}$ | **0.9614**$^{\pm0.0004}$ | 0.9537$^{\pm0.0002}$ | 0.9406$^{\pm0.0087}$ | 0.8311$^{\pm0.0107}$ | 0.9511$^{\pm0.0085}$ | 0.9310$^{\pm0.0120}$ | 0.9580$^{\pm0.0017}$ | 0.9511$^{\pm0.0052}$ | 0.9333$^{\pm0.0202}$ | 0.9522$^{\pm0.0026}$ | 0.9605$^{\pm0.0018}$ |
| glass | 0.7159$^{\pm0.0364}$ | 0.8047$^{\pm0.0711}$ | 0.6097$^{\pm0.0711}$ | 0.6180$^{\pm0.0210}$ | 0.8245$^{\pm0.0290}$ | 0.6348$^{\pm0.0439}$ | 0.6360$^{\pm0.0830}$ | 0.7451$^{\pm0.0790}$ | **0.9449**$^{\pm0.0230}$ | 0.8889$^{\pm0.0276}$ | 0.7607$^{\pm0.0443}$ | 0.8149$^{\pm0.0332}$ | 0.5844$^{\pm0.0861}$ | 0.8276$^{\pm0.0498}$ | 0.8490$^{\pm0.0586}$ |
| ionosphere | 0.9036$^{\pm0.0160}$ | 0.9564$^{\pm0.0134}$ | 0.9651$^{\pm0.0068}$ | 0.7424$^{\pm0.0123}$ | **0.9765**$^{\pm0.0064}$ | 0.9002$^{\pm0.0140}$ | 0.8670$^{\pm0.0302}$ | 0.9494$^{\pm0.0154}$ | 0.9374$^{\pm0.0114}$ | 0.9438$^{\pm0.0182}$ | 0.9638$^{\pm0.0079}$ | 0.9630$^{\pm0.0128}$ | 0.9525$^{\pm0.0155}$ | 0.8742$^{\pm0.0706}$ | 0.9723$^{\pm0.0070}$ |
| mammography | 0.8774$^{\pm0.0060}$ | 0.8390$^{\pm0.0098}$ | 0.8853$^{\pm0.0022}$ | **0.9059**$^{\pm0.0013}$ | 0.8754$^{\pm0.0029}$ | 0.8983$^{\pm0.0023}$ | 0.8645$^{\pm0.0390}$ | 0.7554$^{\pm0.0167}$ | 0.6279$^{\pm0.0149}$ | 0.5885$^{\pm0.0978}$ | 0.8870$^{\pm0.0211}$ | 0.8789$^{\pm0.0057}$ | 0.8414$^{\pm0.0474}$ | 0.8813$^{\pm0.0069}$ | 0.8661$^{\pm0.0099}$ |
| nslkdd | 0.7413$^{\pm0.0301}$ | 0.9536$^{\pm0.0069}$ | 0.5821$^{\pm0.0029}$ | 0.1106$^{\pm0.0003}$ | 0.9659$^{\pm0.0030}$ | 0.5000$^{\pm0.0000}$ | 0.5673$^{\pm0.0704}$ | 0.7408$^{\pm0.0790}$ | 0.9128$^{\pm0.0384}$ | 0.7775$^{\pm0.1697}$ | **0.9712**$^{\pm0.0635}$ | 0.9451$^{\pm0.0317}$ | 0.7603$^{\pm0.1329}$ | 0.6170$^{\pm0.0527}$ | 0.9655$^{\pm0.0088}$ |
| optdigits | 0.8369$^{\pm0.0519}$ | **0.9737**$^{\pm0.0031}$ | 0.6244$^{\pm0.0084}$ | 0.6065$^{\pm0.0044}$ | 0.9458$^{\pm0.0041}$ | 0.5000$^{\pm0.0000}$ | 0.5735$^{\pm0.1342}$ | 0.8303$^{\pm0.0381}$ | 0.9539$^{\pm0.0148}$ | 0.9423$^{\pm0.0192}$ | 0.9447$^{\pm0.0105}$ | 0.8630$^{\pm0.1120}$ | 0.8311$^{\pm0.0547}$ | 0.5317$^{\pm0.0858}$ | 0.8928$^{\pm0.0233}$ |
| pendigits | 0.9667$^{\pm0.0053}$ | 0.9930$^{\pm0.0024}$ | 0.9654$^{\pm0.0029}$ | 0.9276$^{\pm0.0015}$ | **0.9989**$^{\pm0.0003}$ | 0.9424$^{\pm0.0019}$ | 0.9076$^{\pm0.0770}$ | 0.2591$^{\pm0.0503}$ | 0.9809$^{\pm0.0024}$ | 0.9445$^{\pm0.0262}$ | 0.9927$^{\pm0.0068}$ | 0.9652$^{\pm0.0267}$ | 0.9829$^{\pm0.0048}$ | 0.9884$^{\pm0.0064}$ | 0.9942$^{\pm0.0018}$ |
| pima | 0.7290$^{\pm0.0141}$ | 0.7052$^{\pm0.0164}$ | 0.6986$^{\pm0.0154}$ | 0.5909$^{\pm0.0124}$ | **0.7411**$^{\pm0.0124}$ | 0.7106$^{\pm0.0151}$ | 0.6710$^{\pm0.0647}$ | 0.4222$^{\pm0.0257}$ | 0.5610$^{\pm0.0206}$ | 0.6460$^{\pm0.0142}$ | 0.6334$^{\pm0.0049}$ | 0.6567$^{\pm0.0398}$ | 0.6995$^{\pm0.0318}$ | 0.6931$^{\pm0.0146}$ | 0.7099$^{\pm0.0227}$ |
| satellite | 0.7972$^{\pm0.0142}$ | 0.8441$^{\pm0.0062}$ | 0.7568$^{\pm0.0023}$ | 0.5843$^{\pm0.0035}$ | 0.8756$^{\pm0.0036}$ | 0.6614$^{\pm0.0022}$ | 0.6545$^{\pm0.0424}$ | 0.7746$^{\pm0.0093}$ | 0.7774$^{\pm0.0096}$ | 0.8282$^{\pm0.0159}$ | 0.8222$^{\pm0.0070}$ | **0.8971**$^{\pm0.0129}$ | 0.7369$^{\pm0.0196}$ | 0.8594$^{\pm0.0343}$ | 0.8276$^{\pm0.0034}$ |
| satimage-2 | 0.9929$^{\pm0.0016}$ | 0.9960$^{\pm0.0013}$ | 0.9969$^{\pm0.0001}$ | 0.9648$^{\pm0.0004}$ | **0.9990**$^{\pm0.0002}$ | 0.9784$^{\pm0.0003}$ | 0.9674$^{\pm0.0218}$ | 0.9954$^{\pm0.0004}$ | 0.8517$^{\pm0.0101}$ | 0.9902$^{\pm0.0045}$ | 0.9958$^{\pm0.0011}$ | 0.9963$^{\pm0.0022}$ | 0.9980$^{\pm0.0007}$ | 0.9990$^{\pm0.0003}$ | 0.9985$^{\pm0.0004}$ |
| shuttle | 0.9964$^{\pm0.0007}$ | 0.9996$^{\pm0.0002}$ | 0.9964$^{\pm0.0011}$ | 0.9930$^{\pm0.0002}$ | 0.9990$^{\pm0.0002}$ | 0.9939$^{\pm0.0013}$ | 0.9913$^{\pm0.0049}$ | 0.9941$^{\pm0.0022}$ | **0.9997**$^{\pm0.0001}$ | 0.9939$^{\pm0.0046}$ | 0.9986$^{\pm0.0004}$ | 0.9995$^{\pm0.0003}$ | 0.9954$^{\pm0.0034}$ | 0.9986$^{\pm0.0004}$ | 0.9994$^{\pm0.0001}$ |
| thyroid | **0.9900**$^{\pm0.0015}$ | 0.9635$^{\pm0.0133}$ | 0.9840$^{\pm0.0014}$ | 0.9772$^{\pm0.0018}$ | 0.9874$^{\pm0.0014}$ | 0.9827$^{\pm0.0019}$ | 0.9750$^{\pm0.0159}$ | 0.8638$^{\pm0.0212}$ | 0.7999$^{\pm0.0379}$ | 0.7898$^{\pm0.0866}$ | 0.9768$^{\pm0.0051}$ | 0.9204$^{\pm0.0184}$ | 0.9791$^{\pm0.0043}$ | 0.9634$^{\pm0.0164}$ | 0.9800$^{\pm0.0034}$ |
| wbc | 0.9553$^{\pm0.0072}$ | 0.9535$^{\pm0.0054}$ | 0.9571$^{\pm0.0100}$ | 0.8995$^{\pm0.0100}$ | 0.9517$^{\pm0.0058}$ | 0.9527$^{\pm0.0055}$ | 0.9404$^{\pm0.0356}$ | 0.3962$^{\pm0.0591}$ | 0.4783$^{\pm0.0548}$ | 0.7831$^{\pm0.0420}$ | 0.8489$^{\pm0.0743}$ | 0.9459$^{\pm0.0152}$ | 0.9484$^{\pm0.0064}$ | 0.9444$^{\pm0.0161}$ | **0.9606**$^{\pm0.0081}$ |
| wine | 0.8897$^{\pm0.0419}$ | 0.9628$^{\pm0.0146}$ | 0.9593$^{\pm0.0138}$ | 0.7310$^{\pm0.0226}$ | 0.9657$^{\pm0.0130}$ | 0.9243$^{\pm0.0218}$ | 0.8313$^{\pm0.2015}$ | 0.8782$^{\pm0.0397}$ | **0.9708**$^{\pm0.0223}$ | 0.9067$^{\pm0.0698}$ | 0.9637$^{\pm0.0294}$ | 0.9522$^{\pm0.0334}$ | 0.9667$^{\pm0.0140}$ | 0.9582$^{\pm0.0162}$ | 0.9657$^{\pm0.0282}$ |
| Average AUC-PR | 0.8637$^{\pm0.0154}$ | 0.8741$^{\pm0.0113}$ | 0.8497$^{\pm0.0082}$ | 0.7773$^{\pm0.0056}$ | **0.9018**$^{\pm0.0059}$ | 0.8079$^{\pm0.0144}$ | 0.8140$^{\pm0.0511}$ | 0.6868$^{\pm0.0380}$ | 0.7862$^{\pm0.0228}$ | 0.8260$^{\pm0.0396}$ | 0.8829$^{\pm0.0187}$ | 0.8710$^{\pm0.0275}$ | 0.8475$^{\pm0.0294}$ | 0.8425$^{\pm0.0285}$ | 0.8978$^{\pm0.0124}$ |
| Average Rank | 6.7619 | 7.0476 | 6.2381 | 10.2857 | **3.5476** | 9 | 10.8095 | 12.619 | 9.8571 | 9.9524 | 5.9524 | 7.8095 | 8.6667 | 7.4286 | 4.0238 |

Table 9: Ablation study of individual components with standard deviation. In Attn-Enc. and Attn-Dec., ✗ indicates substitution with MLPs, while in Memory it denotes the exclusion of the module.

| Attn-Enc. | Attn-Dec. | Memory | AUC-PR |
|---|---|---|---|
| ✗ | ✗ | ✗ | 0.6184$^{\pm0.0182}$ |
| ✓ | ✗ | ✗ | 0.6672$^{\pm0.0366}$ |
| ✗ | ✓ | ✗ | 0.6458$^{\pm0.0511}$ |
| ✗ | ✗ | ✓ | 0.6304$^{\pm0.0367}$ |
| ✓ | ✓ | ✗ | 0.7083$^{\pm0.0285}$ |
| ✓ | ✗ | ✓ | 0.6726$^{\pm0.0356}$ |
| ✗ | ✓ | ✓ | 0.6937$^{\pm0.0321}$ |
| ✓ | ✓ | ✓ | **0.7128**$^{\pm0.0256}$ |

We further provide the statistical test results in Figure 5. First, LATTE achieves the best average rank overall. While the results indicate that LATTE is statistically on par with top-performing baselines, LATTE demonstrates consistently low standard deviations across datasets. This smaller variance per dataset, coupled with the best average ranking, highlights its robustness and stability, which is critical for practical TAD.

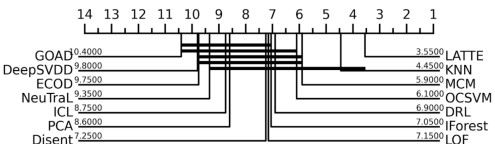

Figure 5: Critical Difference Diagram.

## C SYNTHETIC ANOMALIES

In this section, we provide the full results in Table 2. The details of each anomaly type follow the settings described in (Han et al., 2022).

- **Global Anomalies.** Sampled from a uniform distribution whose range extends beyond the observed minimum and maximum of each feature. The degree of deviation is controlled by $\alpha = 1.1$, producing outliers that lie outside the global feature range.

- **Local Anomalies.** Constructed by first generating normal samples with a GMM (Milligan, 1985) and then amplifying the covariance matrix by a scaling factor ($\alpha = 5$), which yields points that deviate from their neighborhoods.

- **Clustered Anomalies.** Referred to as group anomalies, formed by shifting the mean vectors of normal clusters with a scaling factor ($\alpha = 5$) and then using a scaled GMM to synthesize anomalous clusters separated from normal data.

- **Dependency Anomalies.** Constructed as data points that do not follow the natural dependency structure across features. We employ methods such as Vine Copula (Aas et al., 2009) and kernel density estimation (KDE) (Hastie et al., 2009) to generate synthetic samples in which feature relationships are deliberately removed.

Table 10: Tabular anomaly detection results in global anomalies. The rank indicates the relative AUC-PR performance within each dataset. The best results are shown in **bold** and the second best in underlined.

| Dataset | Machine Learning | | | | | Deep Learning | | | |
|---|---|---|---|---|---|---|---|---|---|
| | IForest | LOF | OCSVM | KNN | PCA | MCM | DRL | Disent | Ours |
| mammography | **0.9975** | 0.8849 | 0.9918 | 0.9932 | 0.9842 | 0.9957 | 0.9974 | 0.9812 | 0.9953 |
| satellite | 0.9980 | **1.0000** | **1.0000** | **1.0000** | 0.9962 | 0.9879 | **1.0000** | 0.9975 | **1.0000** |
| satimage-2 | 0.9313 | **1.0000** | 0.9998 | **1.0000** | 0.8461 | 0.6852 | **1.0000** | 0.9603 | **1.0000** |
| optdigits | **1.0000** | 0.8956 | 0.9963 | 0.9937 | 0.0559 | 0.9957 | 0.9995 | 0.9911 | 0.9945 |
| thyroid | **0.9928** | 0.9648 | 0.9848 | 0.9754 | 0.9644 | 0.9862 | 0.9905 | 0.9681 | 0.9874 |
| cardiotocography | **1.0000** | 0.9995 | 0.9999 | 0.9999 | 0.5526 | 0.9990 | 0.9999 | 0.9983 | 0.9999 |
| ionosphere | 0.9401 | 0.9630 | 0.9988 | **0.9999** | 0.8671 | 0.4902 | 0.9974 | 0.8527 | **0.9999** |
| cardio | **1.0000** | 0.9981 | 0.9996 | 0.9996 | 0.4224 | 0.9996 | **1.0000** | 0.9966 | 0.9994 |
| pima | **0.9951** | 0.9841 | 0.9916 | 0.9924 | 0.9861 | 0.9924 | 0.9731 | 0.9637 | 0.9915 |
| breastw | **0.9973** | 0.9261 | 0.9968 | 0.9941 | 0.9956 | 0.9921 | 0.9897 | 0.9952 | 0.9940 |
| Average AUC | 0.9852 | 0.9616 | 0.9959 | 0.9948 | 0.7671 | 0.9124 | 0.9948 | 0.9705 | **0.9962** |
| Average Rank | **2.8182** | 6.4545 | 3.8182 | 3.7273 | 7.6364 | 6.1818 | 3.9091 | 7.0909 | 3.3636 |

Table 11: Tabular anomaly detection results in local anomalies. The rank indicates the relative AUC-PR performance within each dataset. The best results are shown in **bold** and the second best in underlined.

| Dataset | Machine Learning | | | | | Deep Learning | | | |
|---|---|---|---|---|---|---|---|---|---|
| | IForest | LOF | OCSVM | KNN | PCA | MCM | DRL | Disent | Ours |
| mammography | 0.0856 | **0.7615** | 0.1066 | 0.2874 | 0.0869 | 0.2491 | 0.3120 | 0.3172 | 0.4273 |
| satellite | 0.9391 | 0.9813 | 0.9465 | 0.9713 | 0.8494 | 0.8905 | 0.9488 | 0.8401 | **0.9878** |
| satimage-2 | 0.4925 | 0.7718 | 0.5730 | 0.6188 | 0.4560 | 0.2903 | 0.5199 | 0.4614 | **0.8011** |
| optdigits | 0.4132 | 0.7751 | 0.5906 | 0.7043 | 0.0559 | 0.6471 | 0.7734 | 0.6874 | **0.8411** |
| thyroid | 0.1626 | 0.3276 | 0.1778 | 0.2283 | 0.1603 | 0.1743 | **0.3795** | 0.2407 | 0.3434 |
| cardiotocography | 0.8631 | 0.9640 | 0.9238 | 0.9360 | 0.5190 | 0.9027 | 0.9500 | 0.9209 | **0.9764** |
| ionosphere | 0.7900 | 0.8935 | 0.9341 | 0.9440 | 0.8293 | 0.7079 | 0.9054 | 0.8436 | **0.9561** |
| cardio | 0.6856 | 0.9091 | 0.7903 | 0.8349 | 0.3358 | 0.7638 | 0.8861 | 0.7760 | **0.9315** |
| pima | 0.8712 | 0.9197 | 0.9181 | 0.9166 | 0.8976 | 0.9084 | 0.9110 | 0.8532 | **0.9282** |
| breastw | 0.7022 | 0.7976 | 0.7867 | 0.7638 | 0.7871 | 0.7745 | 0.8061 | **0.8346** | 0.7961 |
| Average AUC | 0.6005 | **0.8101** | 0.6747 | 0.7205 | 0.4977 | 0.6309 | 0.7392 | 0.6775 | 0.7989 |
| Average Rank | 7.8182 | 2.2727 | 5.1818 | 4.1818 | 8.0000 | 7.0909 | 3.3636 | 5.5455 | **1.5455** |

Table 12: Tabular anomaly detection results in clustered anomalies. The rank indicates the relative AUC-PR performance within each dataset. The best results are shown in **bold** and the second best in underlined.

| Dataset | Machine Learning | | | | | Deep Learning | | | |
|---|---|---|---|---|---|---|---|---|---|
| | IForest | LOF | OCSVM | KNN | PCA | MCM | DRL | Disent | Ours |
| mammography | 0.4405 | 0.9661 | 0.9593 | 0.9418 | 0.8291 | 0.6718 | **1.0000** | 0.9835 | 0.9790 |
| satellite | **1.0000** | **1.0000** | **1.0000** | **1.0000** | **1.0000** | 0.9953 | **1.0000** | **1.0000** | **1.0000** |
| satimage-2 | **1.0000** | **1.0000** | **1.0000** | **1.0000** | **1.0000** | 0.8750 | **1.0000** | **1.0000** | **1.0000** |
| optdigits | **0.9867** | 0.9583 | 0.9786 | 0.9338 | 0.0559 | 0.9116 | 0.9380 | 0.8597 | 0.9589 |
| thyroid | **1.0000** | 0.9962 | 0.9937 | 0.997 | 0.9907 | **1.0000** | **1.0000** | 0.9906 | 0.9998 |
| cardiotocography | **1.0000** | **1.0000** | **1.0000** | **1.0000** | 0.5529 | **1.0000** | **1.0000** | **1.0000** | **1.0000** |
| ionosphere | 0.9745 | **1.0000** | **1.0000** | **1.0000** | **1.0000** | **1.0000** | **1.0000** | **1.0000** | **1.0000** |
| cardio | 0.8193 | 0.9815 | 0.9604 | 0.9749 | 0.3956 | 0.9703 | **1.0000** | 0.9192 | 0.9917 |
| pima | **1.0000** | **1.0000** | **1.0000** | **1.0000** | **1.0000** | **1.0000** | **1.0000** | **1.0000** | **1.0000** |
| breastw | **1.0000** | 0.9618 | 0.9986 | 0.9980 | 0.9990 | 0.9956 | 0.9958 | 0.9947 | 0.9979 |
| Average AUC | 0.9221 | 0.9864 | 0.9891 | 0.9846 | 0.7823 | 0.9420 | **0.9934** | 0.9748 | 0.9927 |
| Average Rank | 5.1364 | 4.8636 | 4.1364 | 4.8636 | 6.5455 | 6.2273 | **3.5909** | 5.7727 | 3.8636 |

Table 13: Tabular anomaly detection results in dependency anomalies. The rank indicates the relative AUC-PR performance within each dataset. The best results are shown in **bold** and the second best in underlined.

| Dataset | Machine Learning | | | | | Deep Learning | | | |
|---|---|---|---|---|---|---|---|---|---|
| | IForest | LOF | OCSVM | KNN | PCA | MCM | DRL | Disent | Ours |
| mammography | 0.0757 | 0.7669 | 0.0792 | 0.3725 | 0.0507 | 0.3305 | 0.6619 | 0.1038 | **0.8074** |
| satellite | 0.9443 | 0.9984 | 0.9743 | 0.9951 | 0.4686 | 0.9835 | **0.9996** | 0.5110 | 0.9994 |
| satimage-2 | 0.1785 | 0.9941 | 0.4463 | 0.9374 | 0.0268 | 0.8095 | 0.9914 | 0.0255 | **0.9998** |
| optdigits | 0.1111 | **0.6065** | 0.0874 | 0.4159 | 0.0559 | 0.2884 | 0.4148 | 0.0667 | 0.5941 |
| thyroid | 0.0706 | 0.3037 | 0.0876 | 0.1532 | 0.0741 | 0.2535 | 0.5068 | 0.0636 | **0.5710** |
| cardiotocography | 0.4836 | 0.8377 | 0.4943 | 0.7270 | 0.3684 | 0.7697 | **0.9964** | 0.4168 | 0.9531 |
| ionosphere | 0.6218 | 0.9743 | 0.9040 | 0.9697 | 0.5429 | 0.3894 | 0.9880 | 0.5928 | **0.9930** |
| cardio | 0.2723 | 0.7312 | 0.2956 | 0.5317 | 0.1864 | 0.5847 | **0.9948** | 0.2419 | 0.9033 |
| pima | 0.5813 | 0.6561 | 0.6004 | 0.6318 | 0.5536 | 0.6448 | 0.6627 | 0.5384 | **0.6968** |
| breastw | 0.6288 | 0.6415 | 0.6079 | 0.6104 | 0.6010 | 0.6351 | **0.7752** | 0.6245 | 0.6976 |
| Average AUC | 0.3968 | 0.7510 | 0.4577 | 0.6345 | 0.2928 | 0.5689 | 0.7992 | 0.3185 | **0.8216** |
| Average Rank | 6.8182 | 2.6364 | 6.2727 | 4.5455 | 8.5455 | 4.9091 | 2.0000 | 7.8182 | **1.4545** |

Table 14: Comparison of inference time (seconds).

|  | campaign | shuttle | nslkdd | fraud | census |
|---|---|---|---|---|---|
| Data Statistics | 41188 x 62 | 49097 x 9 | 148517 x 122 | 284807 x 29 | 299285 x 500 |
| Ours | 0.2597 | 0.2706 | 1.0152 | 1.0390 | 4.2573 |
| KNN | 18.531 | 2.9610 | 229.73 | 430.16 | 10286 |
| NPTAD | 119.15 | 16.766 | 96.307 | 34.041 | 1065.2 |
| MCM | 0.2278 | 0.2471 | 0.9693 | 0.8061 | 2.7814 |
| Disent | 0.2895 | 0.1567 | 1.4015 | 1.2582 | 8.1162 |
| ICL | 0.3286 | 0.6969 | 0.9017 | 1.8659 | 4.5346 |
| DRL | 0.1012 | 0.1110 | 0.5117 | 0.7438 | 1.2015 |

## D COMPARISON OF INFERENCE TIME

In this section, we provide inference time comparison with other baselines with data statistics (*i.e.* the number of samples × the number of columns) in Table 14. LATTE is faster than KNN and NPT-AD and comparable to MLP-based baselines, which demonstrates LATTE achieves both effectiveness and efficiency. We can analyze from two complementary perspectives. First, although LATTE employs an attention module to model inter-column dependencies, the use of a latent bottleneck prevents the computational cost from growing quadratically with the number of columns (*i.e.* the number of attention tokens), which makes the model computationally efficient. This enables competitive comparisons with MLP-based models, which are known to be more efficient than attention-based approaches. Second, while KNN is effective but becomes increasingly inefficient as the numbers of samples and columns grow, LATTE adopts the same retrieval concept through a memory bank that stores a small set of prototypical memory vectors instead of all samples. Third, LATTE achieves substantial speeds ups compared to NPT-AD, even if NPT-AD was run on 6 gpus for `NSLKDD`, `Fraud`, and `Census` datasets to accommodate the memory requirements. Unlike NPT-AD, which suffers from both attention between sample and quadratic time complexity with respect to the number of features due to its non-bottleneck structure, LATTE can avoid such high computational costs through its architecture design. Taken together, these design choices allow LATTE to retain the ability of attention to model inter-column dependencies and the retrieval benefits of KNN-style methods, while maintaining both effectiveness and efficiency.

## E ADDITIONAL VISUALIZATIONS

### E.1 RECONSTRUCTION VISUALIZATION

We further analyze the learned representations using a UMAP (McInnes et al., 2018) visualization on `ionosphere`. As illustrated in Figure 6, normal samples remain clustered in consistent regions, whereas abnormal samples are mapped in closer to the normal manifold after reconstruction. These results demonstrate that LATTE effectively regularizes anomalies by reconstructing them toward onto the normal feature space.

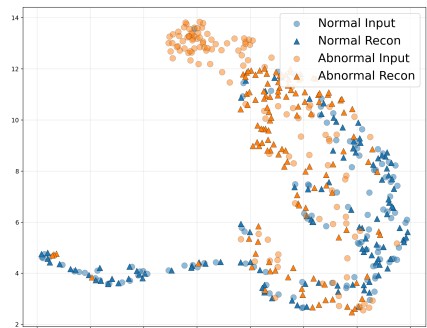

Figure 6: UMAP visualization.

### E.2 TRAINING CURVE

As shown in Figure 7, we analyze the training curves under different encoder-decoder architectures. When MLPs are used in the encoder (Figure 7a), test performance shows large fluctuations and eventually degrades, indicating susceptibility to overfitting. Using an attention based encoder instead (Figure 7b), although performance degradation still persists in the early stages of training, stabilizes training and improves robustness to overfitting. LATTE, applying attention to both the encoder and decoder (Figure 7c), yields the most stable curves and consistently better test metrics. These results indicate

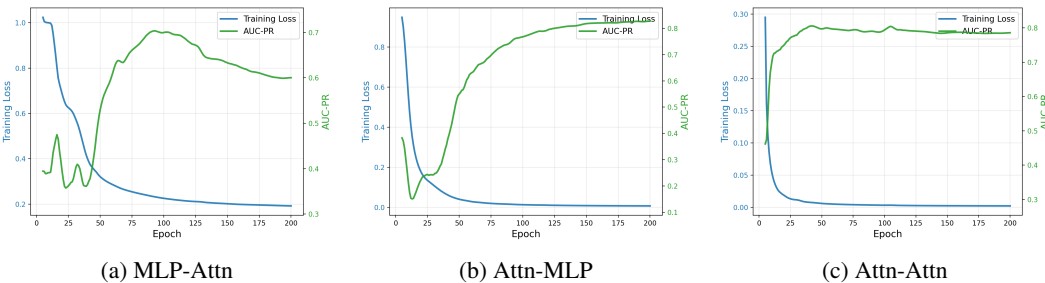

(a) MLP-Attn        (b) Attn-MLP        (c) Attn-Attn

Figure 7: Visualization of training loss and test metric of LATTE under three configurations; sub-captions denote encoder and decoder types in the order encoder-decoder.

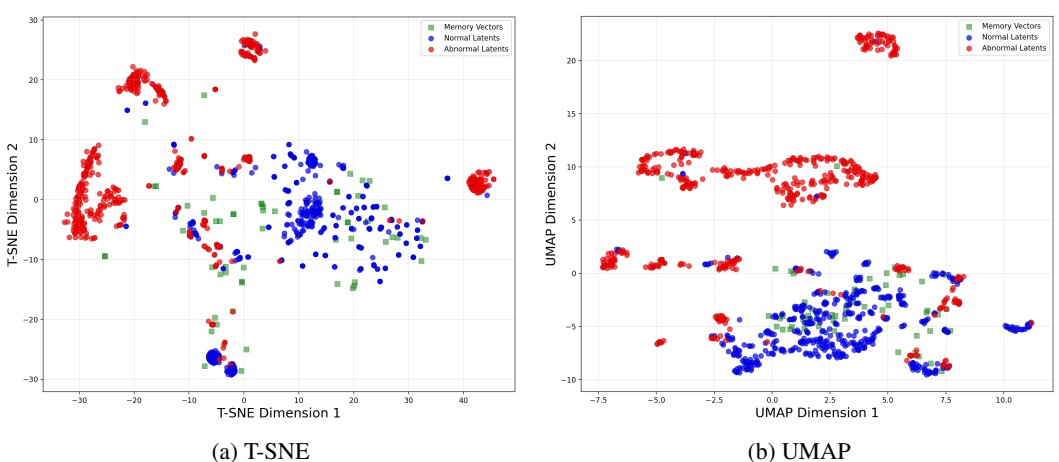

(a) T-SNE              (b) UMAP

Figure 8: Visualization of normal, abnormal latent and memory vectors .

that attention based backbones provide a more reliable architecture for TAD and justifies our design choices.

### E.3 MEMORY VECTOR VISUALIZATION

In this section, we provide memory vector visualization with T-SNE and UMAP on `nslkdd` dataset. Figure 8 shows that normal latent is closely positioned to memory vectors in reduced dimension while abnormal latent is far away from memory vectors.

## F ADDITIONAL BASELINES

In this section, we compare LATTE with Thimonier et al. (2024b), which resembles NPT-AD but relies on reconstructing augmented versions of the input via attention between samples. Table 15 shows that LATTE consistently outperforms this retrieval-augmented baseline, validating the effectiveness of introducing our learned memory bank over retrieval from the entire training set.

Table 15: Performance comparison between LATTE and Thimonier et al. (2024b) The best results are shown in **bold**.

| Models | br | card | cardt | glass | ion | mamm | pend | pima | wbc | wine | Avg |
|---|---|---|---|---|---|---|---|---|---|---|---|
| Thimonier et al. (2024b) | 0.5184 | 0.1753 | **0.6984** | 0.0804 | 0.8907 | 0.0454 | 0.66336 | 0.5174 | 0.7546 | 0.1469 | 0.4491 |
| **Ours** | **0.9844** | **0.8442** | 0.6811 | **0.2909** | **0.9772** | **0.4196** | **0.8679** | **0.6986** | **0.7837** | **0.8266** | **0.7374** |

