# OpenReview forum: "Tabular Anomaly Detection via Reconstruction with Attention-Based Bottleneck"
_ICLR.cc/2026/Conference — Submitted to ICLR 2026_

### Official Review · Reviewer_LM4S · 2025-10-21

**Soundness:** 4
**Presentation:** 3
**Contribution:** 3
**Rating:** 6
**Confidence:** 5

**Summary:**

The present work proposes a novel attention-based and memory augmented anomaly detection method for tabular data. Their approach focuses on the one-class classification setting whereby the training set only contains *normal* samples, and the overall aim is to characterize this *normal* distribution to identify out-of-distribution samples in inference.

The proposed method, coined LATTE, implicitely combines inter-feature dependencies through using attention-based architectures, and inter-sample relations by training a memory bank used to augment the latent representation. LATTE is a reconstruction-based method that consists in training a model to reconstruct a sample, in an autoencoder-like approach, and uses the reconstruction error as the anomaly score in inference; the higher the error the less likely to belong to the normal distribution the tested sample is.

The authors test their method on 20 datasets built as a combination of the ADBench and ODD benchmarks. Their experimental results demonstrate the relevance of their method as they obtain competitive performance based on the AUPRC metric and average rank.
A thorough ablation study demonstrate the relevance of each components in LATTE.

**Strengths:**

**S1**: The paper is well-written, easy to follow and properly structured.

**S2**: Approach is novel, well motivated and offers strong performance.

**S3**: Experiments are rigorous and compare to most relevant methods found in the literature.

**S4**: Some ablations studies and figures are particularly relevant, e.g. fig 3 and 4.

**S5:** Work is fully reproductible.

**Weaknesses:**

**W1**: Some relevant references are missing. In particular, the authors mention on several occasions how very few attention based methods can be found in the literature. Moreover, they rely on a memory-bank to augment their reconstruction-based method. To our knowledge, [1] proposes a relatively similar approach in spirit. While demonstrating significantly poorer performance than the proposed method, [1] puts forward an attention-based method that involves a retrieval module akin to the memory bank in spirit. It might be worth mentioning in the related works, or including it in the benchmark.

**W2**: Ablation studies, although interesting, do not mention any statistical testing between set-ups. Given the small variation between obtained metrics in some cases, e.g. Table 3 and 4 or Figure 2.

**W3**: Some relevant experimental details are missing from the paper (see questions).

**W4**: No discussion on complexity/training time can be found in the paper.

[1] Retrieval Augmented Deep Anomaly Detection for Tabular Data. *Hugo Thimonier, Fabrice Popineau, Arpad Rimmel and Bich-Liên Doan*. CIKM 2024.

**Questions:**

Some minor typos were found in the submitted manuscript:
- line 181: "(...) tokens Z as a effective (...)" -> an.
- line 365: "We conduct how different (...)".

**Q1**: In light of recent work [3], while the authors mention how AUPRC has been considered as a relevant metric for evaluating AD methods, it might also be interesting to consider some alternative metrics, e.g., AUROC and threshold-dependent metrics like the F1-Score as previously used in the literature [2, 4, 5]. This should be quick, as the code provided in the supplementary material shows that they have been computed.

**Q2**: As mentioned in **W3**, it might be interesting to:
- (i) mention what criterion was used to stop training,
- (ii) display training curves with and without the different additions mentioned in Table 3.

**Q3**: As mentioned in **W2**, no statistical testing is mentioned in the ablation, while they are critical to motivate an approach. We encourage the authors to augment section 4 with statistical tests to further demonstrate the relevance of their approach.

**Q4**: While the authors rely on T-SNE and provide a very informative visualization in Figure 4, [6] shows how T-SNE is very sensitive to hyperparameters, and any representations can be obtained with a well-chosen set of parameters. Could the authors provide a similar visualization using a different dimension reduction method, e.g., UMAP?

**Q5**: A possible additional visualization could also be helpful to understand the underlying mechanisms driving LATTE: could the authors provide a representation (using T-SNE/UMAP) of the chosen vectors in the memory bank for normal samples vs anomaly samples, to see how close to the sample of interest they are? I expect the chosen vectors from the memory bank to be scattered in the representation space for anomalies, while being close for normal samples.

Given the strengths, weaknesses, and interrogations I have, I lean towards accepting the paper. I am open to increasing my score should the authors address my concerns/interrogations.

[2] DRL: Decomposed Representation Learning for Tabular Anomaly Detection. *Hangting Ye, He Zhao, Wei Fan, Mingyuan Zhou, Dan dan Guo, Yi Chang*. ICLR 2024.

[3] A Closer Look at AUROC and AUPRC under Class Imbalance. *Matthew McDermott, Haoran Zhang, Lasse Hansen, Giovanni Angelotti and Jack Gallifant*. NeurIPS 2024.

[4] MCM: Masked Cell Modeling for Anomaly Detection in Tabular Data. *Jiaxin Yin, Yuanyuan Qiao, Zitang Zhou, Xiangchao Wang, Jie Yang*. ICLR 2024.

[5] Anomaly Detection for Tabular Data with Internal Contrastive Learning. *Tom Shenkar, Lior Wolf*. ICLR 2022

[6] How to Use t-SNE Effectively", Wattenberg, et al., "Distill, 2016. http://doi.org/10.23915/distill.00002

---

> ### Author Response · Authors · 2025-11-21
> **Rebuttal by Authors [1/2]**
>
> Dear Reviewer LM4S,
>
> We sincerely appreciate your helpful feedback to improve the qualities of our paper.  We have provided a detailed response to each of your concerns.
>
> ---
>
> >**[W1]** Some relevant references are missing.
>
> **[Response]**
> As you suggested, we append [1] in related works and include the comparison between LATTE and [1] in Appendix of our revision. We here evaluate the method [1] on 10 small benchmarks since it requires the quadratic computation cost with respect to the number of training samples, which makes comparison infeasible. As shown in the below table, LATTE consistently outperforms [1], validating the effectiveness of our learned memory bank over retrieval from the entire training set.
>
> |  | br | card | cardt | glass | ion | mamm | pend | pima | wbc | wine | AVG |
> | :---- | :---- | :---- | :---- | :---- | :---- | :---- | :---- | :---- | :---- | :---- | :---- |
> | \[1\] | 0.5184 | 0.1753 | **0.6984** | 0.0804 | 0.8907 | 0.0454 | 0.6636 | 0.5174 | 0.7546 | 0.1469 | 0.4491 |
> | Ours | **0.9844** | **0.8442** | 0.6811 | **0.2909** | **0.9772** | **0.4196** | **0.8679** | **0.6986** | **0.7837** | **0.8266** | **0.7374** |
>
> [1] Retrieval Augmented Deep Anomaly Detection for Tabular Data.
>
> ---
>
> >**[W4]** No discussion on complexity/training time can be found in the paper.
>
> **[Response]**
> Thank you for your suggestion. We have included the computational cost analysis in Table 14.
>
> ---
>
> >**[Q1]** It might also be interesting to consider some alternative metrics, e.g., AUROC and threshold-dependent metrics like the F1-Score.
>
> **[Response]**
> Following your suggestion, we include alternative metrics (AUROC and F1-Score) in our revision. Please refer to Appendix B.
>
> ---
>
> >**[Q2-1 / W3]** It might be interesting to mention what criterion was used to stop training.
>
> **[Response]**
> Following a popular baseline MCM, we trained our model with 200 epochs without early stopping. We simply used the last-epoch checkpoint for evaluation.
>
> [1] MCM: Masked Cell Modeling for Anomaly Detection in Tabular Data, ICLR 2024
>
>
> ---
>
> >**[Q2-2 / W3]** It might be interesting to display training curves with and without the different additions mentioned in Table 3.
>
> **[Response]**
> As you suggested, we have included the training curves and test metrics for the ablation study components in Figure 7. Please note that test metrics are reported solely for analysis and were not used as criteria for early stopping.
>
> We observe that attention consistently facilitates stable training dynamics. Using a MLP-based Encoder exhibits significant fluctuations in test metrics, eventually leading to overfitting. In contrast, incorporating attention into the encoder improves robustness by enabling better feature extraction, despite some performance instability in the early training stages. LATTE, applying attention in both the encoder and decoder, yields the most stable training curve and robust performance, These results demonstrate that attention-based backbones can provide a more reliable architectural foundation for TAD compared to MLP-based designs.
>
> ---
>
> >**[Q3 / W2]** No statistical testing is mentioned in the ablation. Encourage the authors to augment section 4 with statistical tests to further demonstrate the relevance of their approach.
>
> **[Response]**
> We would like to mention that standard deviations for all experiments are reported in Table 6, 7 and 8. To further address your concern, we also perform a statistical test using our main experiments (Figure 5).
>
> As shown in Figure 5, LATTE achieves the best average rank overall. While the analysis indicates that LATTE is statistically on par with top-performing baselines, we would like to emphasize that LATTE demonstrates consistently low standard deviations across datasets (as detailed in Appendix 2). This smaller variance per dataset, coupled with the best average ranking, highlights its robustness and stability, which is critical for practical TAD.
>
> ---
>
> >**[Q4]** Could the authors provide a similar visualization using a different dimension reduction method, e.g., UMAP?
>
> **[Response]**
> We appreciate the valuable suggestion and have included the UMAP visualization in the Figure 6 of our revised manuscript. The visualization shows that the reconstructions of abnormal samples closely moves to the normal data distribution, which is consistent with the initial findings in section 4.4. This further indicates that LATTE effectively maps abnormal samples onto the normal manifold as intended.

---

> ### Author Response · Authors · 2025-11-21
> **Rebuttal by Authors [2/2]**
>
> >**[Q5]** Could the authors provide a representation (using T-SNE/UMAP) of the chosen vectors in the memory bank for normal samples vs anomaly samples, to see how close to the sample of interest they are? I expect the chosen vectors from the memory bank to be scattered in the representation space for anomalies, while being close for normal samples.
>
> **[Response]**
> We appreciate your valuable advice for better understanding our architecture. We included the t-SNE and UMAP visualization of the latent representations and retrieved memory vectors on $\texttt{nslkdd}$ dataset included in Figure 8.
>
> As you expected, the visualization demonstrates that the latent representations of normal samples are positioned closely to the retrieved memory vectors. In contrast, the latent representations of abnormal samples are located significantly far away from the memory vectors. This distinct separation confirms that while our memory bank effectively retrieves prototypical patterns for normal data, it cannot find suitable prototypes for anomalies, thereby amplifying the discrepancy in the latent space.

---

> > ### Comment · Reviewer_LM4S · 2025-11-25
> > **Answer to Rebuttal**
> >
> > Dear Authors,
> >
> > Thank you for answering my questions and trying to address the point raised.
> >
> > Still a few minor points:
> >
> > 1. Fig 7: Authors mention how they rely on the approach from [1] to select number of training epochs. This seems as quite odd: the required number of epochs to train a model is highly dependent on the complexity of the dataset and the model's capacity. Also, there is absolutely no reason for [1]'s method to require a same number of epoch for convergence as LATTE. Always selecting 200 epochs to train a model could either lead to (i) overfitting, (ii) underfitting or (iii) be a simple waste of compute.
> > All three sub figures displayed in Fig. 7 support this as the training loss converges after less than 100 epochs for all tested set-ups. In particular, *Attn-Attn* displays loss convergence after only 50 epochs.
> > We encourage the authors to propose a stopping criterion (e.g. no improvement on training loss for $k$ consecutive epochs).
> >
> > 2. A few issues related to the reference section:
> >
> > First, one should try and avoid citing preprints when the mentioned papers have been published, e.g.:
> > - Chen Qiu, Timo Pfrommer, Marius Kloft, Stephan Mandt, and Maja Rudolph. *Neural transformation learning for deep anomaly detection beyond images* -> accepted to ICML 2021.
> > - Léo Grinsztajn, Edouard Oyallon, and Gaël Varoquaux. *Why do tree-based models still outperform deep learning on tabular data?* -> accepted at NeurIPS 2022.
> > - Hugo Thimonier, Fabrice Popineau, Arpad Rimmel, and Bich-Liên Doan. *Beyond individual input for deep anomaly detection on tabular data* -> ICML 2024
> > - Jianan Ye, Zhaorui Tan, Yijie Hu, Xi Yang, Guangliang Cheng, and Kaizhu Huang. *Disentangling tabular data towards better one-class anomaly detection* -> AAAI 2025
> > - Alexey Dosovitskiy, Lucas Beyer, Alexander Kolesnikov, Dirk Weissenborn, Xiaohua Zhai, Thomas Unterthiner, Mostafa Dehghani, Matthias Minderer, Georg Heigold, Sylvain Gelly, Jakob Uszkoreit, and Neil Houlsby. *An image is worth 16x16 words: Transformers for image recognition at scale* -> ICLR 2021
> > - Ashish Vaswani, Noam Shazeer, Niki Parmar, Jakob Uszkoreit, Llion Jones, Aidan N. Gomez, Lukasz Kaiser, and Illia Polosukhin. *Attention is all you need* -> NeurIPS 2017
> >
> > Second, a reference is mentioned twice:
> > - Xin Huang, Ashish Khetan, Milan Cvitkovic, and Zohar Karnin. Tabtransformer: Tabular data modeling using contextual embeddings. arXiv, 2012.06678v1, 2020a
> >
> > 1] MCM: Masked Cell Modeling for Anomaly Detection in Tabular Data, ICLR 2024

---

> > > ### Author Response · Authors · 2025-11-27
> > > **Additional response to Reviewer LM4S**
> > >
> > > Thank you very much for your thoughtful follow-up questions and for actively engaging in the discussion.
> > >
> > > > **[Q1]** Authors mention how they rely on the approach from [1] to select the number of training epochs. This seems as quite odd: the required number of epochs to train a model is highly dependent on the complexity of the dataset and the model’s capacity. Also, there is absolutely no reason for [1]’s method to require the same number of epochs for convergence as LATTE. Always selecting 200 epochs to train a model could either lead to (i) overfitting, (ii) underfitting or (iii) be a simple waste of compute. All three sub figures displayed in Fig.7 support this as the training loss convergences after less than 100 epochs for all tested set-ups. In particular, Attn-Attn displays loss convergence after only 50 epochs. We encourage the authors to propose stopping criteria (e.g., no improvement on training loss for k consecutive epochs).
> > >
> > > **[Response]**
> > >
> > > Thank you for raising this important point regarding training stopping criteria.
> > >
> > > Following your suggestion, we have implemented an early stopping strategy with a patience of 10 for small scale dataset and 20 for large scale dataset. Notably, this adaptive criterion slightly improves AUC-PR and significantly reduces training time for several datasets while maintaining performance. For example, training on the *wbc* dataset converged at 100 epochs, whereas the *fraud* dataset required approximately 300 epochs, both yielding an improvement. These results demonstrate that an early stopping effectively guides the model to a dataset-specific convergence point, ensuring robust detection accuracy and efficiency.
> > >
> > > |  | ar | br | cam | car | cat | cen | fraud | gla | ion | mam |
> > > | :---- | :---- | :---- | :---- | :---- | :---- | :---- | :---- | :---- | :---- | :---- |
> > > | **epoch 200** | 0.6113 | 0.9844 | 0.5105 | 0.8442 | 0.6811 | 0.2474 | 0.7240 | 0.2909 | 0.9772 | **0.4196** |
> > > | **early stopping** | **0.6116** | **0.9845** | **0.5114** | **0.8445** | **0.6819** | **0.2496** | **0.7273** | **0.2925** | **0.9785** | 0.4154 |
> > >
> > >
> > > |  | nsl | opt | pen | pim | sat | sai | shu | thy | wbc | wine | AVG |
> > > | :---- | :---- | :---- | :---- | :---- | :---- | :---- | :---- | :---- | :---- | :---- | :---- |
> > > | **epoch 200** | **0.9755** | **0.2204** | 0.8679 | 0.6986 | 0.8657 | 0.9747 | **0.9889** | **0.7566** | 0.7837 | **0.8260** | 0.7124 |
> > > | **early stopping** | 0.9744 | 0.2170 | **0.8698** | **0.6990** | **0.8661** | **0.9748** | 0.9883 | 0.7547 | **0,7887** | 0.8254 | **0.7128** |
> > >
> > >
> > > We would like to note that, in practical scenarios, a labeled validation set might be unavailable due to the absence of labeled anomalies, which prevents monitoring metrics for early stopping. Therefore, achieving robustness to training duration without reliance on a validation set is important in unsupervised tabular anomaly detection; hence, we simply adopted a fixed strategy of 200 epochs with an exponential learning rate scheduler and did not tune the strategy.
> > >
> > > To further assess LATTE’s robustness to training duration and its susceptibility to overfitting, we evaluate LATTE across diverse epochs (ranging from 50 to 500). As shown in the table below, LATTE exhibits consistent performance after 150 epochs. The negligible performance variance across extended epochs demonstrates that LATTE is robust against overfitting, even when trained beyond convergence. However, the early stopping strategy remains preferable; it yields marginal performance gains while optimizing training time by preventing redundant computations.
> > >
> > >
> > > | Epochs | 50 | 100 | 150 | 200 | 250 | 300 | 350 | 400 | 450 | 500 |
> > > | :---- | :---- | :---- | :---- | :---- | :---- | :---- | :---- | :---- | :---- | :---- |
> > > | **AUC-PR** | 0.7014 | 0.7099 | 0.7122 | 0.7124 | 0.7124 | **0.7126** | 0.7125 | 0.7124 | 0.7124 | 0.7124 |
> > >
> > >
> > > ---
> > > > **[Q2]** A few issues related to the reference section.
> > >
> > > **[Response]**
> > > We appreciate the reviewer’s detailed review. We have resolved issues about the reference section on our revision.

---

> > > > ### Comment · Reviewer_LM4S · 2025-11-27
> > > > **Response to authors on additional experiments**
> > > >
> > > > Dear Authors,
> > > >
> > > > Thank you for conducting those additional experiments.
> > > >
> > > > Regarding the following comment from the authors "*We would like to note that, in practical scenarios, a labeled validation set might be unavailable due to the absence of labeled anomalies, which prevents monitoring metrics for early stopping. Therefore, achieving robustness to training duration without reliance on a validation set is important in unsupervised tabular anomaly detection; hence, we simply adopted a fixed strategy of 200 epochs with an exponential learning rate scheduler and did not tune the strategy.*" -> apologies as it was a poor choice of word from our end. I should not have refered as "early stopping" to describe stopping training when the training loss stops improving for $k$ consecutive epochs.
> > > >
> > > > As described in my previous comment, I was considering a stopping criterion based on how the training loss evolves over time. Can you confirm that the table shown in your response is the result of stopping training when the training loss does improve for 10 (or 20) consecutive epochs?

---

> > > > > ### Author Response · Authors · 2025-11-27
> > > > > **Additional response to Reviewer LM4S**
> > > > >
> > > > > > **[Q1]** As described in my previous comment, I was considering a stopping criterion based on how the training loss evolves over time. Can you confirm that the table shown in your response is the result of stopping training when the training loss does improve for 10 (or 20) consecutive epochs?
> > > > >
> > > > > **[Response]**
> > > > > Thank you for your request for clarification.
> > > > >
> > > > > Yes, we confirm that the table shown in our previous response is the result of stopping training when the training loss does not improve for 10 (or 20) consecutive epochs.

---

> > > > > > ### Comment · Reviewer_LM4S · 2025-11-27
> > > > > > **Final response to authors**
> > > > > >
> > > > > > Dear Authors,
> > > > > >
> > > > > > Thank you again for the clarifications and additional experiments.
> > > > > > Following the comment at the end of my initial review, I am happy to increase my score.
> > > > > >
> > > > > > Nevertheless, I strongly urge the authors to replace their original results obtained from training systematically with a number of epochs set to 200, to the ones obtained with the stopping strategy just discussed. This would be a more robust methodology.

---

> > > > > > > ### Author Response · Authors · 2025-11-28
> > > > > > > **Answer to Final Response**
> > > > > > >
> > > > > > > Thank you for your active engagement.
> > > > > > >
> > > > > > > We have updated the revision to reflect the points discussed so far.
> > > > > > >
> > > > > > > Please feel free to let us know if you have any further questions.

---

### Official Review · Reviewer_Q6HB · 2025-10-29

**Soundness:** 2
**Presentation:** 2
**Contribution:** 2
**Rating:** 2
**Confidence:** 4

**Summary:**

The paper proposes LATTE, a reconstruction-based framework for unsupervised tabular anomaly detection (TAD). LATTE introduces two key architectural components: (1) an attention-based bottleneck that replaces conventional MLP encoders/decoders to better model inter-column dependencies, and (2) a learnable memory bank inspired by k-nearest neighbors (KNN) to store prototypical normal patterns and amplify reconstruction errors for anomalies. The method is evaluated on 20 standard TAD benchmarks.

**Strengths:**

•	The paper is well-structured and clearly explains the architecture and experimental setup. Figures and equations are effectively used to illustrate the model design.
•	The implementation details are thorough, and the ablation studies (e.g., on attention components, memory bank size, temperature, and query design) demonstrate careful empirical validation of design decisions.

**Weaknesses:**

•	While the paper claims that attention mechanisms better capture inter-column dependencies, it does not clearly articulate what specific limitations of MLPs in TAD this addresses, nor does it provide empirical or theoretical evidence (e.g., attention maps, feature interaction analysis) showing that MLPs fundamentally fail to model such dependencies. The use of attention feels more like a design choice than a solution to a well-defined problem. A stronger motivation would link the architectural design to known failure modes of existing methods (e.g., on dependency-type anomalies).
•	The paper asserts that LATTE “consistently outperforms” state-of-the-art methods based on average AUC-PR and rank. However, a closer inspection of Table 1 reveals that LATTE achieves the best result on only 3 out of 20 datasets, while being outperformed by baselines like KNN, Disent, or MCM on many others (e.g., breastw, cardio, pendigits, wine). The average metrics can mask substantial performance variance, and the claim of consistent superiority is therefore overstated.
•	The individual components—attention bottlenecks and memory banks—are well-established in other domains, and thus the novelty is somewhat limited. A more nuanced discussion of architectural novelty versus engineering integration would be warranted.

**Questions:**

Attention mechanisms and memory addressing typically increase computational cost compared to MLPs or KNN. Could the authors report inference time relative to baselines?

---

> ### Author Response · Authors · 2025-11-21
> **Rebuttal by Authors [1/2]**
>
> Dear Reviewer Q6HB,
>
> We truly appreciate your thoughtful review. We have carefully considered your points and addressed them individually in the following.
>
> ---
>
> >**[W1]** While the paper claims that attention mechanisms better capture inter-column dependencies, it does not clearly articulate what specific limitations of MLPs in TAD this addresses, nor does it provide empirical or theoretical evidence (e.g., attention maps, feature interaction analysis) showing that MLPs fundamentally fail to model such dependencies. A stronger motivation would link the architectural design to known failure modes of existing methods (e.g., on dependency-type anomalies).
>
> **[Response]**
> Thank you for your intuition and constructive comment.
>
> We believe that an attention is more suitable for TAD than MLPs since it explicitly models relationships between columns. Following your suggestion, to further validate the effectiveness of attention-based bottleneck compared to MLPs, we report a component ablation study specifically on the dependency anomaly type, where anomaly does not follow feature correlations that normal samples must have.
>
> The pure MLP architecture yields a critically low AUC-PR of 0.3005, confirming that standard MLPs fundamentally fail to model complex inter-column dependencies. Specifically, they cannot capture the correlation violations when the marginal feature distributions appear normal, which is the defining characteristic of dependency anomalies.
>
> In contrast, LATTE achieves an AUC-PR of 0.8216, representing a +0.5211 absolute improvement of AUCPR over the variant 1. Furthermore, replacing either the MLP encoder or MLP decoder with an attention-based one results in AUC-PR of 0.7251 or 0.6278, respectively, which are large improvements compared to variant 1. These results demonstrate that attention is superior to MLP for capturing complex dependencies between columns.
>
> |  | Encoder type | Decoder type | AVG-AUCPR |
> | :---- | :---- | :---- | :---- |
> | Variant 1 | MLP | MLP | 0.3005 |
> | Variant 2 | MLP | Attention | 0.6278 |
> | Variant 3 | Attention | MLP | 0.7251 |
> | Ours | Attention | Attention | **0.8216** |
>
> For additional results with other anomaly types (e.g., local, global and cluster), see Table 4 in our revision.
>
> ---
>
> >**[W2]** The average metrics can mask substantial performance variance, and the claim of consistent superiority is therefore overstated.
>
> **[Response]**
> We believe that our framework is more effective and robust than others since we achieve SOTA performance with the same hyperparameters across all benchmarks unlike previous works (e.g., MCM[1] and Disent[2]) that used different hyperparameter choices per dataset. To provide a direct comparison under similar conditions, we report the results of LATTE with per-dataset hyperparameters. As shown below (1st rank and 2nd rank are represented by **bold** and *italic*, repectively), LATTE achieves 1st rank on 6 datasets and 2nd rank on 5 datasets out of 20 benchmarks.
>
> However, we gracefully argue that the fixed configurations approach presented in our manuscript is more aligned with the nature of unsupervised TAD. Therefore, we believe that LATTE’s ability to deliver high performance without dataset-specific configurations highlights its strong effectiveness and robustness.
>
>
> | ar | br | cam | car | cat | cen | fra | gla | ion | mam |
> | :---- | :---- | :---- | :---- | :---- | :---- | :---- | :---- | :---- | :---- |
> |0.6161 | 0.9893 | **0.5210** | **0.8565** | *0.7248* | **0.2564** | *0.7282* | 0.3273 | **0.9816** | 0.4400 |
>
> | nsl | opt | pen | pim | sat | sai | shu | thy | wbc | wine | AVG |
> | :---- | :---- | :---- | :---- | :---- | :---- | :---- | :---- | :---- | :---- | :---- |
> | *0.9779* | 0.3709 | *0.9102* | 0.7030 | 0.8730 | *0.9773* | 0.9917 | 0.7786 | **0.8083** | **0.8869** | **0.7360** |
>
> [1] MCM: Masked Cell Modeling for Anomaly Detection in Tabular Data, ICLR 2024
>
> [2] Disentangling Tabular Data Towards Better One-Class Anomaly Detection, AAAI 2025
>
> ---
>
> >**[W3]** The individual components—attention bottlenecks and memory banks—are well-established in other domains, and thus the novelty is somewhat limited.
>
> **[Response]**
> We would like to emphasize that our novelty lies in **bridging the gap between deep architectures and traditional ML methods**. Specifically, we revisit architectural designs and integrate principles from traditional ML methods into deep learning models, which are nontrivial advances in the field of tabular domain and tabular anomaly detection. Although our individual components are inspired by existing methods, the superior performance of LATTE validates that these architectural integrations are highly promising for advancing the current state of TAD. We believe that our overall framework is novel enough and would be impactful for guiding future architectural designs in TAD.

---

> ### Author Response · Authors · 2025-11-21
> **Rebuttal by Authors [2/2]**
>
> >**[Q1]** Attention mechanisms and memory addressing typically increase computational cost compared to MLPs or KNN. Could the authors report inference time relative to baselines?
>
> **[Response]**
> Our attention mechanisms and memory addressing do not increase the computational cost much because we employ an attention-based bottleneck architecture. This design does not suffer from quadratic computational complexity $O(F^2)$ with respect to the number of features, $F$, by projecting the $F$ features onto a small number of latent tokens, $M (M \ll F)$. Moreover, unlike KNN that requires computation with the entire dataset, our learnable memory bank maintains only a compact set of $K$ representative prototype vectors. Consequently, LATTE can be much faster than KNN and comparable to MLP-based baselines.
>
> Following your suggestions, we report the comparison of inference time between LATTE, KNN, and MLP-based baselines, MCM, ICL, Disent, and DRL. In below table, the row "Data Statistics" denotes the dataset dimension (number of sampels $\times$ the number of columns), and each cell represents inference time in seconds. The results demonstrate that while KNN scales poorly with large-scale datasets, LATTE achieves inference speeds comparable to MLP-based baselines, highlighting the efficiency of the architectural choices.
>
> |  | campaign | shuttle | nslkdd | fraud | census |
> | :---- | :---- | :---- | :---- | :---- | :---- |
> | Data Statistics  | 41188 x 62 | 49097 x 9 | 148517 x 122 | 284807 x 29 | 299285 x 500 |
> | Ours | 0.2597 | 0.2706 | 1.0152 | 1.0390 | 4.2573 |
> | KNN | 18.531 | 2.9610 | 229.73 | 430.16 | 10286 |
> | MCM | 0.2278 | 0.2471 | 0.9693 | 0.8061 | 2.7814 |
> | Disent | 0.2895 | 0.1567 | 1.4015 | 1.2582 | 8.1162 |
> | ICL | 0.3286 | 0.6969 | 0.9017 | 1.8659 | 4.5346 |
> | DRL | 0.1012 | 0.1110 | 0.5117 | 0.7438 | 1.2015 |
>
> We have included this analysis in our revision (see Table 14).

---

> ### Comment · Area_Chair_pbjU · 2025-11-28
> **Please reply to the authors' rebuttal**
>
> Dear Reviewer,
>
> The authors have provided their rebuttal. Please reply to it before the rebuttal period ends. Thanks!
>
> Best regards,
>
> AC

---

### Official Review · Reviewer_xpHw · 2025-10-31

**Soundness:** 2
**Presentation:** 2
**Contribution:** 2
**Rating:** 2
**Confidence:** 4

**Summary:**

This paper focuses on the one class tabular anomaly detection task. The proposed method follows the reconstruction-based method, and incorporate attention-based method to capture the normal distribution. An additional latent memory bank is designed to incorporate the global normal characteristics.

**Strengths:**

S1: The studied problem is important.

S2: The paper structure is clear and easy to follow.

S3: The compared baselines are extensive.

**Weaknesses:**

W1: Towards the novelty of this paper. (1) The paper claimed one of their core motivations is inspired by KNN, but this is confused. The fundamental logic of KNN-based anomaly detection is to leverage the local relationships between samples (**input space**), where a normal sample is assumed to be closer to its neighbors in the training set than an anomaly would be. In contrast, the mechanism of the proposed latent memory bank operates differently. In this paper, latent memory bank assumes that normal sample representation Z, compared to anomalous representation,  is more close to the distribution of learned global vectors M in the bank, thus each normal sample represnetation can be better replaced as the combination of M (**latent space**), thus could be better decoded to reconstruct input sample. This approach appears less aligned with the sample-wise relational reasoning of KNN and more reminiscent of the concept in DRL, which assumes that normal representation could be better reprensentd by a linear combination of global random vectors (**latent space**). Could the authors clarify the specific inspiration drawn from KNN?

(2) The second point is critical, as it substantially impacts the perceived novelty of the paper. The paper positions the combination of attention mechanisms and KNN-inspired concepts as a key contribution. However, this specific combination seems to have been recently and directly explored by the prior work NPT-AD [1], which incorporates attention across both features and samples to model their interactions for sample reconstruction. The absence of a comparison with this highly relevant and important baseline significantly weakens the novelty claims. Could the authors explain the unique advantages and methodological distinctions of their approach compared to NPT-AD?

[1] Beyond Individual Input for Deep Anomaly Detection on Tabular Data. ICML 2024.

W2: In Line 164, the authors mentioned that 'be interpreted as the positional embedding of the i-th column'. But in tabular domain, features are order-agnostic, thus positional embedding is not appropriate for this type of data.

W3: The encoder-decoder architecture uses the low-rank based projection, which is similar to the SetTransformer [2]. Why not directly using SetTransformer? What is the performance benefits compared to the more complex design?

[2] Set Transformer: A Framework for Attention-based Permutation-Invariant Neural Networks. ICML 2019.

W4: Based on W2, why using such low-rank projection? Whai if the latent dimension of Z is the same as input X?

W5: In Eq.6, why bi is from the encoder, rather than be the new learnable parameters? It seems that it can limit the representation learning ability in such auto encoder based architecture. And it is different from the autoencoder family. Is there any specific reason?

Based on W1 to W5, the rationale behind the methodology requires thorough elaboration.

W6: Is there any theoretical analysis?

**Questions:**

Please see weaknesses above.

---

> ### Author Response · Authors · 2025-11-21
> **Rebuttal by Authors [1/3]**
>
> Dear Reviewer xpHw,
>
> We sincerely appreciate your helpful feedback and insightful comments. We address your concerns one by one.
>
> ---
>
> >**[W1-1]** The fundamental logic of KNN-based anomaly detection is to leverage the local relationships between samples (input space), where a normal sample is assumed to be closer to its neighbors in the training set than an anomaly would be. The proposed approach appears less aligned with the sample-wise relational reasoning of KNN and more reminiscent of the concept in DRL. Could the authors clarify the specific inspiration drawn from KNN?
>
> **[Response]**
> Thank you for your insightful comment regarding the relationship between KNN, DRL, and LATTE.
>
> (LATTE vs KNN) Our memory module is well aligned with the retrieval logic of KNN. Whereas KNN retrieves the top-K most similar samples from the training set, our module retrieves a weighted combination of prototypical normal latents that represent the distribution of normal training data. Although our approach does not compare against every individual normal sample, retrieving from a compact set of representative latent vectors provides an efficient and effective approximation of KNN-based relational reasoning. In this sense, our method retains the core intuition of KNN while operating in a learned latent space that captures more expressive and semantically structured relationships among normal samples.
>
> (LATTE vs DRL) We would like to emphasize that LATTE and DRL diverge fundamentally in how they define and model normality. The key distinction lies in the nature of the latent basis: LATTE utilizes learnable memory vectors to model the actual data distribution, whereas DRL relies on fixed, randomized basis vectors to impose a structural constraint.
>
> Specifically, LATTE draws inspiration from KNN’s retrieval mechanism, employing learnable prototypes optimized to capture the manifold of normal data. It assumes that normal data should be reconstructed via **patterns observed during training**. In contrast, DRL projects data onto a fixed, randomized subspace, assuming that normal data can be decomposed into global basis vectors in latent space, which necessitates additional loss terms to enforce this constraint.
>
> [1] DRL: Decomposed Representation Learning for Tabular Anomaly Detection, ICLR 2025
>
> ---

---

> > ### Comment · Reviewer_xpHw · 2025-11-28
> >
> > Thanks the authors for the response. However, there are still concerns unsolved.
> >
> > **For W1:** The authors state that memory module is well aligned with the retrieval logic of KNN, rather than DRL.  The difference between the memory module and DRL is whether the latent space vectors is fixed or learned. It seems that  the difference between the memory module and KNN locates in (i) latent space/observation space, (ii) this paper utilizes the combination of all vectors (which is more like topic model), but KNN utilizes the local relationships between samples (not all samples). To my opinion, the proposed method is not more like KNN. However, it is OK that the authors are entitled to maintain their position that this is more similar to KNN.
> >
> > If the proposed method is more like KNN, it should be noted that such a view implicitly acknowledges the importance of dependencies among normal samples—a point already emphasized in NPT-AD. Therefore, NPT-AD should serves as one important baseline method, and authors should justify the relation and difference between them in the main text. Only additional results (and this is even not full dataset comparison of NPT-AD) in the appendix is not enough.
> >
> > **For W2:** I did not agree with this point. It is well acknowledged by many papers [1][2] that positional embedding should be dropped to satisfy the feature order-agnostic nature.
> >
> > [1] Making Pre-trained Language Models Great on Tabular Prediction. ICLR 2024.
> >
> > [2] Towards Cross-Table Masked Pretraining for Web Data Mining. WWW 2024.
> >
> > **For W3:** The point I emphasized earlier is that the idea of using the low-rank based projection in encoder-decoder architecture is similar to SetTransformer.
> >
> > **For W4:** The emperical evidence is needed.
> >
> > **For W6:** As I already indicated, the theoretical analysis is not enough.

---

> > > ### Author Response · Authors · 2025-12-01
> > > **Additional response to Reviewer xpHw [2/2]**
> > >
> > > >**[W3]** The point I emphasized earlier is that the idea of using the low-rank based
> > > projection in encoder-decoder architecture is similar to SetTransformer.
> > >
> > > **[Response]**
> > > We appreciate the reviewer for pointing out the connection to SetTransformer.
> > >
> > > The critical distinction between LATTE and SetTransformer lies in the **decoding mechanism**. In SetTransformer, Induced Set Attention Block (ISAB) decodes induced latents by projecting induced latents to the original feature embedding space. In the context of reconstruction-based anomaly detection, using the input itself as a query allows the model to bypass the information bottleneck, potentially leading to a trivial identity mapping.
> > >
> > > In contrast, LATTE employs a query-based decoder where queries are created by the sum of  learnable global embeddings and positional embeddings ($q_{global}+b_i$), which are independent of the specific input feature values. This design choice enforces a strict bottleneck and enables the model to filter out anomalies; hence, our architectural choice is better suited for the TAD task than the ISAB of SetTransformer.
> > >
> > > To empirically validate this, we conducted an ablation study by replacing our encoder-decoder backbone with SetTransformer. As shown in the table below, this variant significantly underperforms LATTE. These results demonstrate that our architectural choice, decoupling the decoder query from the input,  is a key factor in LATTE’s superior performance.
> > >
> > > |  | glass | optdigits | pendigits | wbc | wine | AVG (20 datasets) |
> > > | :---- | :---- | :---- | :---- | :---- | :---- | :---- |
> > > | LATTE | 0.2925 | 0.2170 | 0.8696 | 0.7887 | 0.8254 | 0.7128 |
> > > | SetTransformer​​ | 0.1590 | 0.0598 | 0.3458 | 0.7694 | 0.6618 | 0.6254 |
> > >
> > > ---
> > >
> > > >**[W4]**  Empirical evidence is needed.
> > >
> > > **[Response]**
> > > To address the reviewer’s concern, we conducted an ablation study varying the number of latent tokens. As shown in the table below, setting the number of latent tokens, $M$, equal to the number of input columns ,$F$, results in suboptimal performance.
> > >
> > > |  | glass | optdigits | pendigits | wbc | wine | AVG (20 datasets) |
> > > | :---- | :---- | :---- | :---- | :---- | :---- | :---- |
> > > | LATTE | 0.2925 | 0.2170 | 0.8698 | 0.7887 | 0.8254 | 0.7128 |
> > > | Variant (No Bottleneck)​​ | 0.1953 | 0.1099 | 0.6682 | 0.7677 | 0.7037 | 0.6737 |
> > >
> > > When the latent capacity is sufficiently large (i.e., $M \approx F$), the model tends to learn a trivial identity mapping to minimize reconstruction loss, rather than capturing meaningful inter-column dependencies. The encoder creates a shortcut that bypasses the information bottleneck, and the model solely relies on the memory module for anomaly detection. The empirical evidence validates the necessity of our attention-based bottleneck design (i.e., $M \ll F$).

---

> ### Author Response · Authors · 2025-11-21
> **Rebuttal by Authors [2/3]**
>
> >**[W1-2]** Could the authors explain the unique advantages and methodological distinctions of their approach compared to NPT-AD?
>
> **[Response]**
> LATTE differs from NPT-AD[1] in the following aspects:
> 1. Key idea:
>     - NPT-AD is a masked modeling approach. It detects anomalies by masking parts of an input and reconstructing them by **modeling dependencies between training samples explicitly**.
>     - LATTE is a full reconstruction approach. It compresses a single instance into a latent representation and reconstructs it by **retrieving learned normal prototypes** from a memory bank.
>
> 2. Architecture:
>     - To reconstruct masked features of a single test instance, **NPT-AD requires the entire training set as an input**, which causes a very high computation cost to perform attention across all the training samples and their columns.
>     - **LATTE only takes a single instance as input** and computes its (compressed) latent using our attention-based bottleneck structure with the fixed-size, learnable memory bank.
>
> 3. Performance:
>     - Due to these architectural differences, LATTE outperforms NPT-AD with significantly faster inference speed. Unlike NPT-AD, which suffers from **quadratic complexity with respect to the number of training samples** due to its attention usage, LATTE avoids these high computational costs by performing retrieval from a fixed-size, trainable memory bank; thus, achieving time complexity **independent of dataset size**.
>
> 4. Empirical Result:
>     - To validate the superiority of LATTE compared to NPT-AD, we conduct comparisons on 10 benchmarks since NPT-AD requires high computational cost on large-scale datasets. LATTE consistently outperforms   NPT-AD and achieves substantial speedups ranging from 1.94$\times$ to 1962.47$\times$ .
>
> **Performance Comparison between LATTE and NPT-AD**
> |  | br | card | cardt | glass | ion | mamm | pend | pima | wbc | wine | AVG-AUCPR  |
> | :---- | :---- | :---- | :---- | :---- | :---- | :---- | :---- | :---- | :---- | :---- | :---- |
> | NPT-AD | 0.9781 | 0.8374 | **0.7420** | **0.2964** | 0.8907 | 0.4007 | 0.7044 | 0.6847 | 0.7395 | 0.7588 | 0.7032 |
> | Ours | **0.9844** | **0.8442** | 0.6811 | 0.2909 | **0.9772** | **0.4196** | **0.8679** | **0.6986** | **0.7837** | **0.8266** | **0.7374** |
>
> **Inference time (seconds) comparison between LATTE and NPT-AD.**
> |  | br | card | cardt | glass | ion | mamm | pend | pima | wbc | wine |
> | :---- | :---- | :---- | :---- | :---- | :---- | :---- | :---- | :---- | :---- | :---- |
> | NPT-AD | 0.0529 | 7.3323 | 10.229 | 0.0486 | 57.108 | 0.7233 | 5.2381 | 0.0698 | 41.143 | 0.0747 |
> | Ours | 0.0272  | 0.0343 | 0.0410 | 0.0244 | 0.0291 | 0.0457 | 0.0735 | 0.0343 | 0.0239 | 0.0286 |
> | speedup (x) | 1.94 | 213.77 | 249.49 | 1.99 | 1962.47 | 15.83 | 71.27 | 2.03 | 1721.46 | 2.61 |
>
> We have included these experiments in our revision (see Table 15 and 16).
>
> ---
>
> **[W2]** In the tabular domain, features are order-agnostic, thus positional embedding is not appropriate for this type of data.
>
> **[Response]**
> Thank you for pointing this out. While we agree that features are order-agnostic, in tabular data, features within a single row follow strictly column-aware structures. For example, if columns 0 and 1 correspond to “Age” and “Salary” respectively, then obviously, two vectors like [20, 80] and [80, 20] have entirely different meanings. Since self-attention is permutation-invariant, it treats all input tokens identically without explicit indicators. Thus, we employ positional embedding to serve as column identifiers.
>
> ---
>
> [1] Beyond Individual Input for Deep Anomaly Detection on Tabular Data. ICML 2024.

---

> ### Author Response · Authors · 2025-11-21
> **Rebuttal by Authors [3/3]**
>
> >**[W2]** In the tabular domain, features are order-agnostic, thus positional embedding is not appropriate for this type of data.
>
> **[Response]**
> Thank you for pointing this out. While we agree that features are order-agnostic, in tabular data, features within a single row follow strictly column-aware structures. For example, if columns 0 and 1 correspond to “Age” and “Salary” respectively, then obviously, two vectors like [20, 80] and [80, 20] have entirely different meanings. Since self-attention is permutation-invariant, it treats all input tokens identically without explicit indicators. Thus, we employ positional embedding to serve as column identifiers.
>
> ---
>
> >**[W3]** Why not directly using SetTransformer? What are the performance benefits compared to the more complex design?
>
> **[Response]**
> We argue that the SetTransformer[2] is not suitable for TAD due to its nature of permutation invariance. As mentioned in the response of **[W2]**, tabular features possess strict column-aware structures. SetTransformer is specifically designed to be permutation invariant, treating inputs as an unordered set. Consequently, SetTransformer treats semantically different vectors, such as [20, 80] and [80, 20], identical inputs. Since distinguishing these patterns is fundamental to TAD, using a set-based architecture without positional information is unsuitable for this task.
>
> [2] Set Transformer: A Framework for Attention-based Permutation-Invariant Neural Networks. ICML 2019.
>
>
> ---
>
> >**[W4]** Why using such low-rank projection? What if the latent dimension of $\mathbf{Z}$ is the same as input X?
>
> **[Response]**
> Without low-rank projection (bottleneck structure in LATTE), the model might be trained to learn a trivial identity function, leading to perfect reconstruction for all inputs, regardless of whether they are normal or abnormal samples. Specifically, if the number of latent vectors $N$ is the same as the number of input columns $F$, then the model might learn to simply copy input features to their corresponding latent tokens, preventing the model from capturing compact and useful information for TAD.
>
> ---
>
> >**[W5]** Why is $b_i$ from the Encoder, rather than be the new learnable parameters?
>
> **[Response]**
> We reuse encoder’s $b_i$ in the decoder to make learning easier and more stable by enforcing a single, shared representation for each column. $b_i$ is the column specific embedding that encodes the identity of column $i$, and sharing it ensures consistent column semantics across encoder and decoder. If the decoder had separate column embeddings, the model would first have to learn to align two different representations of the same column. Using the same $b_i$ ​avoids this unnecessary alignment step and removes redundant parameters.
>
> |  | Enc-Dec $b\_i$ shared | AVG-AUCPR |
> | :---- | :---- | :---- |
> | Variant 1 | O | **0.7124** |
> | Variant 2 | X | 0.7089 |
>
> ---
>
> **[W6]** Is there any theoretical Analysis?
>
> **[Response]**
> We appreciate the reviewer's comment and want to inform you that we derived our insights primarily from empirical results rather than theoretical analysis. Even if the theoretical analysis can further strengthen the contribution of our work, we believe that the lack of such proofs would not undervalue our work. Many popular baselines in TAD (e.g., MCM, ICL) have been impactful work with their empirical effectiveness, rather than formal and theoretical proofs.

---

> ### Comment · Area_Chair_pbjU · 2025-11-28
> **Please reply to the authors' rebuttal**
>
> Dear Reviewer,
>
> The authors have provided their rebuttal. Please reply to it before the rebuttal period ends. Thanks!
>
> Best regards,
>
> AC

---

> ### Author Response · Authors · 2025-12-01
> **Additional response to Reviewer xpHw [1/2]**
>
> >**[W1-1]** The authors state that the memory module is well aligned with the retrieval logic of KNN, rather than DRL. The difference between the memory module and DRL is whether the latent space vector is fixed or learned. It seems that the difference between the memory module and KNN locates in (i) latent space/observation space, (ii) this paper utilizes the combination of all vectors (which is more like topic model), but KNN utilizes the local relationships between samples (not all samples). In my opinion, the proposed method is not more like KNN. However, it is OK that the authors are entitled to maintain their position that this is more similar to KNN.
>
> **[Response]**
> We would like to maintain that our proposed method aligns with the core idea of KNN.
>
> In the context of deep learning, extending KNN to the latent space is a well-established approach to capture semantic similarities. For example, ProtoKNN [1] utilizes a KNN classifier by comparing latents with prototypes, and MemAE [2] retrieves prototypes as nearest neighbors and uses linear combination of them to reconstruct normal patterns. These works demonstrate that retrieving prototypes based on similarity in the latent space can be viewed as a KNN-based methodology.
>
> In this sense, we believe that the distinctions mentioned in (i) latent space/observation space and (ii) the combination of all vectors do not deviate from the fundamental principles of KNN, as our module essentially utilizes the nearest neighbor prototypes.
>
>
> [1] This Looks Like It Rather Than That: ProtoKNN For Similarity-Based Classifiers, ICLR 2023.
>
> [2] Memorizing Normality to Detect Anomaly: Memory-augmented Deep Autoencoder for Unsupervised Anomaly Detection, ICCV.
>
> ---
>
> >**[W1-2]** If the proposed method is more like KNN, it should be noted that such a view implicitly acknowledges the importance of dependencies among normal samples—a point already emphasized in NPT-AD. Therefore, NPT-AD should serve as one important baseline method, and authors should justify the relation and difference between them in the main text. Only additional results (and this is not even a full dataset comparison of NPT-AD) in the appendix is not enough.
>
> **[Response]**
>
> As you suggested, we have included the comparison of our method with NPT-AD in the main paper.
>
>
> ---
>
> >**[W2]**  I did not agree with this point. It is well acknowledged by many papers [1][2] that positional embedding should be dropped to satisfy the feature order-agnostic nature.
>
> **[Response]**
> We would like to emphasize that these works preserve feature identifiability by incorporating column name embeddings. Crucially, while [1] and [2] did not use explicit positional embeddings, this is not for satisfying the feature order-agnostic nature.
>
> Specifically, [1] constructs feature embeddings by integrating column name embeddings with value embeddings (e.g., $E_i = E^{name}_i  \otimes E^{value}_i $). Although [1] state that they “remove position encoding on value vectors”, they compensate for this by allowing the model to distinguish features via their semantic name embeddings. Similarly, while [2] mentions discarding positional encoding “to accommodate the permutation invariance of tabular data”, this design choice is tailored for their cross-table learning setting, not aligned with our setting at all. Furthermore, they employed tokenization of column names to generate feature embeddings, which likewise ensures that the model can discriminate between different columns.
>
> To further explore the effectiveness of positional embeddings, we conducted an ablation study by removing them from the feature embedding. As shown in the table below, removing positional embeddings significantly degrades the detection performance. These results validate that the model requires positional embeddings to effectively distinguish between different features.
>
> |  | glass | optdigits | pendigits | wbc | wine | AVG (on 20 datasets) |
> | :---- | :---- | :---- | :---- | :---- | :---- | :---- |
> | LATTE | 0.2925 | 0.2170 | 0.8698 | 0.7887 | 0.8254 | 0.7128 |
> | LATTE (no pos embedding) | 0.1565 | 0.0620 | 0.3507 | 0.7685 | 0.6563 | 0.6266 |
>
> In conclusion, the key lies in the capability to distinguish between features. While [1] and [2] utilize column names for this, our approach aligns with other recent studies [3][4] by employing explicit positional embeddings to ensure feature awareness.
>
>
> [1] Making Pre-trained Language Models Great on Tabular Prediction. ICLR 2024.
>
> [2] Towards Cross-Table Masked Pretraining for Web Data Mining. WWW 2024.
>
> [3] Accurate predictions on small data with a tabular foundation model, Nature 2025
>
> [4] Revisiting Deep Learning Models for Tabular Data, ICLR 2021

---

### Official Review · Reviewer_MPiY · 2025-11-01

**Soundness:** 2
**Presentation:** 3
**Contribution:** 2
**Rating:** 2
**Confidence:** 4

**Summary:**

This paper proposes LATTE, a reconstruction-based framework for tabular anomaly detection (TAD). The authors motivate their work by observing that many deep TAD methods rely on simple MLP architectures and are often outperformed by traditional methods like KNN . LATTE is an autoencoder-style model that replaces the standard MLP bottleneck with an attention-based architecture (using cross- and self-attention blocks) . Additionally, inspired by KNN, the authors introduce a learnable memory bank that intercepts the latent tokens and replaces them with prototypical normal patterns via an attention-based "memory addressing" step . The model is trained to minimize reconstruction error on normal data. The authors report SOTA performance on 20 TAD benchmarks.

**Strengths:**

- The paper correctly identifies that many prior reconstruction-based TAD methods use overly simplistic MLP autoencoders. By replacing this with a more powerful attention-based (i.e., Transformer-style) encoder and decoder, the model is inherently better equipped to capture the complex inter-column dependencies crucial for TAD.
- The model achieves strong average AUC-PR on a wide range of real-world and synthetic datasets (Tables 1 & 2), demonstrating the empirical effectiveness of using a modern attention architecture for this task.

**Weaknesses:**

1. The paper's framing is highly problematic. It heavily relies on the "inspiration" from KNN's strong performance. However, the proposed "Memory Module" is implemented as a learnable, attention-based lookup (Eq. 9), which is fundamentally different from the non-parametric, distance-based retrieval of k-nearest neighbors. This "KNN-inspired" narrative feels forced and does not accurately reflect the model's Transformer-based mechanism.
2. The paper's primary architectural novelty is the "Latent Memory Bank". However, the ablation study in Table 3 shows that removing this memory module entirely results in a performance drop of only 0.0041 AUC-PR (0.7124 vs. 0.7083). This indicates the memory module contributes less than 1% of the model's total performance. The vast majority (99.4%) of the model's effectiveness comes from simply using a superior Attention-based Autoencoder backbone (Attn-Enc + Attn-Dec).
3. When the negligible memory module is set aside, the paper's core architecture is an attention-based autoencoder (cross-attention encoder, self-attention bottleneck, cross-attention decoder) . This is a known architectural pattern (e.g., a variant of the Perceiver model). The paper's SOTA results are therefore unsurprising—it simply demonstrates that a powerful, modern Transformer-based backbone outperforms the older MLP-based backbones used by prior work. This is an expected engineering result, not a fundamental conceptual advance.

**Questions:**

Re: Weakness #1 & #2: Given that the ablation in Table 3 shows the Memory Module contributes only 0.6% (0.7124 vs 0.7083) to the final AUC-PR, how can the authors justify this component as a core contribution? Furthermore, how is this learnable attention-based lookup (Eq. 9) functionally analogous to the non-parametric, distance-based retrieval of KNN, as claimed in the motivation?

Re: Weakness #3: The paper's primary performance gain (Table 3) comes from replacing the MLP Encoder/Decoder with Attention-based ones. Is the paper's main takeaway simply that "Transformer-based autoencoders are better than MLP-based autoencoders for tabular data"? If so, how does this differentiate from existing work on deep learning for tabular data (e.g., TabTransformer, FT-Transformer) ?

---

> ### Author Response · Authors · 2025-11-21
> **Rebuttal by Authors**
>
> Dear Reviewer MPiY,
>
> We would like to thank you for your valuable comments, which have improved the quality of our work. We have provided a detailed response to each of your concerns.
>
> ---
>
> >**[Q1-1 / W2]** How can the authors justify the memory module as a core contribution?
>
> **[Response]**
> To justify the contribution of our memory module, we conduct comprehensive ablation experiments covering all combinations of components, as shown in the table below. These results show that **the memory module consistently provides performance improvements**, even when our attention-based encoder or decoder is not used. This demonstrates that the memory module is equally essential to the encoder and decoder architectures for achieving strong performance in tabular anomaly detection. We have included these additional results in the revision (see Table 3).
> | **Encoder type** | **Decoder type** | **Memory** | **AUC-PR** |
> | :---- | :---- | :---- | :---- |
> | MLP | MLP | x | 0.6184 |
> | MLP | MLP | v | 0.6304 |
> | Attn | MLP | x | 0.6672 |
> | Attn | MLP | v | 0.6726 |
> | MLP | Attn | x | 0.6458 |
> | MLP | Attn | v | 0.6937 |
> | Attn | Attn | x | 0.7083 |
> | Attn | Attn | v | **0.7124** |
>
> ---
>
> >**[Q1-2 / W1]** How is this learnable attention-based lookup (Eq. 9) functionally analogous to the non-parametric, distance-based retrieval of KNN?
>
> **[Response]**
> Given the memory vectors $\{\mathbf{m}_i\}$, the attention-based lookup (Eq. 9) can be viewed as **a soft (i.e., weighted) form of non-parametric retrieval based on cosine distance**. When the temperature scaling hyperparameter $\tau$ is very small, the lookup retrieves only the most similar memory vector, which corresponds to 1-NN. More generally, the lookup aggregates multiple memory vectors similar to the input latent $\mathbf{z}$ according to their cosine distances. Therefore, at inference time, our memory module and KNN are functionally analogous in terms of retrieval behavior.
>
> During training, the retrieved memory vectors are used to reconstruct normal samples, so they gradually become similar to normal latent vectors. Since the number of memory vectors is smaller than the dataset size, the memory vectors learn to represent clusters of normal latent vectors. Therefore, after training, our memory module performs a soft retrieval operation on the latent space.
>
> We believe that our memory module effectively integrates the KNN-style retrieval mechanism into a learnable reconstruction-based AD framework. We kindly ask the reviewer to reconsider the contribution of our memory module. We have also revised the manuscript to further clarify this point (see Section 3.2).
>
> ---
>
>
> >**[Q2 / W3]** Is the paper's main takeaway that “Transformer-based autoencoders are better than MLP-based autoencoders for tabular data”? If so, how does this differentiate from existing work on deep learning for tabular data (e.g., TabTransformer, FT-Transformer)?
>
> **[Response]** :
> Our main takeaway is two-fold: (i) architectural choices in tabular anomaly detection (TAD) are crucial yet remain underexplored, and (ii) the core principles of traditional methods can be effectively incorporated into modern deep architectures. In our work, we demonstrate that both (i) and (ii) consistently improves the detection performance (see Table 3).
>
> We also note that our framework differs fundamentally from deep tabular architectures such as TabTransformer [1] and FT-Transformer [2]. These models cannot be directly employed for reconstruction-based AD, which requires an explicit bottleneck latent space to compress and reconstruct inputs so that anomalies yield large reconstruction errors. In contrast, our architecture incorporates an attention-based bottleneck and a memory module that explicitly shape the latent space to amplify reconstruction errors for anomalous samples.
>
>
> [1] TabTransformer: Tabular Data Modeling Using Contextual Embeddings.
>
> [2] Revisiting Deep Learning Models for Tabular Data. NeurIPS 2021.

---

> ### Comment · Area_Chair_pbjU · 2025-11-28
> **Please reply to the authors' rebuttal**
>
> Dear Reviewer,
>
> The authors have provided their rebuttal. Please reply to it before the rebuttal period ends. Thanks!
>
> Best regards,
>
> AC

---

### Author Response · Authors · 2025-11-21
**General Response**

Dear reviewers and AC,

We genuinely appreciate your valuable time and effort spent reviewing our manuscript.

As reviewers highlighted, we propose a novel method (Reviewer LM4S) in Tabular Anomaly Detection (TAD) literature with strong empirical results (Reviewer MPiY, LM4S) and extensive experiments (Reviewer MPiY, xpHw, Q6HB, LM4S) focusing on an important problem (Reviewer MPiy, xpHw). Our method is well motivated (Reviewer LM4S) and well-written (Reviewer xpHw, Q6HB, LM4S).

We appreciate your constructive comments on our manuscript. In response to the comments, we have carefully revised and enhanced the manuscript with the following additional discussions and experiments:

- Clarify descriptions and statements throughout our manuscript
- Add more detailed ablation study (Table 3)
- Add component ablation on synthetic anomaly types (Table 4)
- Add Main table with different metrics such as AUROC and F-1 Score (Table 7, 8)
- Add statistical test (Figure 5)
- Add inference time analysis (Table 14)
- Add comparison of ours and NPT-AD [1] (Table 15, 16)
- Add the reference [2] (Related work) and comparison with ours (Table 17)
- Add visualization analysis (Figure 6, 7, 8)

These updates are temporarily highlighted in “blue” for your convenience to check.

We hope our response and revision sincerely address all the reviewers’ concerns.

Thank you for your effort.


Best regards,
Authors

[1] Beyond Individual Input for Deep Anomaly Detection on Tabular Data. ICML 2024.

[2] Retrieval Augmented Deep Anomaly Detection for Tabular Data. Hugo Thimonier, Fabrice Popineau, Arpad Rimmel and Bich-Liên Doan. CIKM 2024.

---

### Author Response · Authors · 2025-11-27
**A Gentle Reminder to AC and Reviewers**

Dear AC and Reviewers,

We hope this message finds you well. We are writing to kindly follow up regarding our rebuttal.

We have made a sincere effort to address the reviewers' concerns through detailed clarifications, additional experiments, and further analysis. We believe these additions meaningfully strengthen our paper and help clarify important points.

If there are any remaining concerns or unresolved issues, please feel free to let us know. We would be happy to provide further clarification.

Thank you again for your time and valuable feedback.

Best regards,
Authors

---

### Meta-Review · Area_Chair_eTbP · 2026-01-02

**Summary:**

The authors propose LATTE, a reconstruction-based framework using an attention bottleneck and a memory bank. However, the majority of the reviewers (3 out of 4) recommend a rejection. The consensus among reviewers MPiY, xpHw, and Q6HB is that the paper’s primary performance gains are from moving from MLP to standard Attention backbones. Another critical point is that the marginal use of the "Memory Module," which was shown by the ablation study, about less than 1% to the total performance (0.7124 vs. 0.7083 AUC-PR).

**Reviewer Concerns:**

The authors provided comparisons with the NPT-AD baseline, included runtime and inference speed analysis. They also added statistical significance tests, and incorporated alternative metrics (AUROC/F1-score) as requested by Reviewer LM4S. They also clarified their training strategy and stopping criteria.

However, Reviewers MPiY and xpHw remain unconvinced by the justification for the Memory Module as a "core contribution" given its negligible impact on performance. For the KNN part, some reviewers argues that the mechanism is more aligned with existing decomposed representation learning of existing works. Additionally, the limited architectural novelty is the problem. The method is essentially a variant of a Perceiver-style model applied to tabular data, which has not shown a unique, fundamental insight for the Tabular Anomaly Detection (TAD).

**Reviewer Scores:**

LM4S score: 6 -> 8
Q6HB: 2
MPiY: 2
xpHw: 2

---

### Decision · Program_Chairs · 2026-01-26

Reject